# In-Context Learning of Stochastic Differential Equations with Foundation Inference Models

**Patrick Seifner[1,2], Kostadin Cvejoski[1,3], David Berghaus[1,3]**
**César Ojeda[4] & Ramsés J. Sánchez[1,2,3]**
Lamarr Institute[1], University of Bonn[2], Fraunhofer IAIS[3] & University of Potsdam[4]
seifner@cs.uni-bonn.de, sanchez@cs.uni-bonn.de

## Abstract

Stochastic differential equations (SDEs) describe dynamical systems where deterministic flows, governed by a drift function, are superimposed with random fluctuations, dictated by a diffusion function. The accurate estimation (*or discovery*) of these functions from data is a central problem in machine learning, with wide application across the natural and social sciences. Yet current solutions either rely heavily on prior knowledge of the dynamics or involve intricate training procedures. We introduce `FIM-SDE` (Foundation Inference Model for SDEs), a pretrained recognition model that delivers accurate *in-context* (or zero-shot) estimation of the drift and diffusion functions of *low-dimensional* SDEs, from noisy time series data, and allows rapid *finetuning* to target datasets. Leveraging concepts from amortized inference and neural operators, we (pre)train `FIM-SDE` in a supervised fashion to map a large set of noisy, discretely observed SDE paths onto the space of drift and diffusion functions. We demonstrate that `FIM-SDE` achieves robust *in-context* function estimation across a wide range of synthetic and real-world processes — from canonical SDE systems (*e.g.* double-well dynamics or weakly perturbed Lorenz attractors) to stock price recordings and oil-price and wind-speed fluctuations — while matching the performance of symbolic, Gaussian process and Neural SDE baselines trained on the target datasets. When *finetuned* to the target processes, we show that `FIM-SDE` consistently outperforms all these baselines.

Our pretrained model, repository and tutorials are available online[1].

## 1 Introduction

Stochastic differential equations (SDEs) govern dynamical systems featuring deterministic flows superimposed with random fluctuations. The deterministic components of these systems are dictated by a *drift function*, and can be used to model the slow, tractable or accessible degrees of freedom of dynamic phenomena. The random components are instead controlled by a *diffusion function*, and can be used to represent fast or intractable degrees of freedom. Such a description in terms of tractable versus intractable, slow versus fast, is abstract enough to be approximately valid across disciplines and observation scales, for phenomena in and out of equilibrium. However, the central obstacle limiting its applicability to many real-world scenarios lies in the accurate estimation of the drift and diffusion functions that best characterize a collection of empirical observations. In other words, the problem of *SDE discovery from data*, whose solution is a key step toward the automation of scientific discovery. Yet current machine learning approaches either rely heavily on prior knowledge of the underlying dynamics or involve intricate training schemes.

Likely inspired by the proliferation and advancement of foundation models in natural language processing and time series forecasting, there has been a recent shift toward (pre)training neural

---

[1]https://fim4science.github.io/OpenFIM/intro.html

39th Conference on Neural Information Processing Systems (NeurIPS 2025).

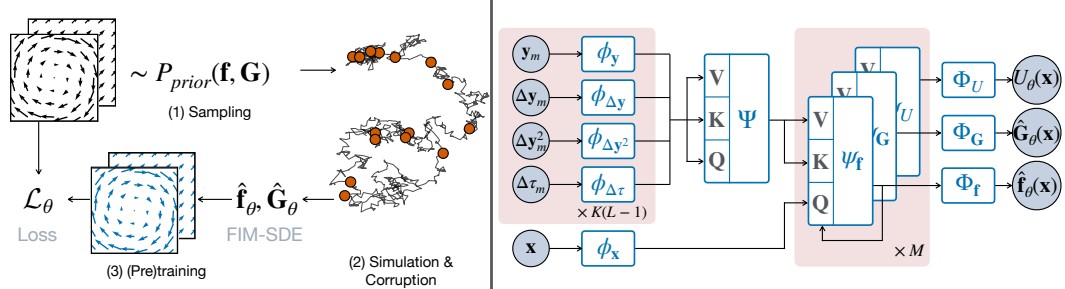

Figure 1: *Left:* Three-step pretraining strategy for SDE discovery problem: (1) sample target drift and diffusion $\mathbf{f}, \mathbf{G}$ from the prior distribution $p_{prior}$; (2) simulate and corrupt SDE paths, with the circles denoting noisy observations; and (3) compute neural estimators $\hat{\mathbf{f}}_\theta, \hat{\mathbf{G}}_\theta$, and match them to the target functions. *Right:* Foundation Inference Model for SDEs (schematic representation). The input (context) consists of $K(L-1)$ tuples of the form $(\mathbf{y}, \Delta\mathbf{y}, \Delta\mathbf{y}^2, \Delta\tau)$ that are projected by the linear $\phi$ layers. The result is processed by a Transformer encoder $\Psi$ that returns the *Context Matrix* (Eq. 7). This matrix is used as keys and values to $M$ *Functional Attention* layers: $\psi_{\mathbf{f}}, \psi_{\mathbf{G}}$ and $\psi_U$. The input queries are the (embedded) location $\mathbf{x}$ where we evaluate the estimated functions.

network models on large, *synthetic datasets* to perform *in-context* (or *zero-shot*) estimation of functions governing the evolution of *unseen* dynamical systems. Examples include models that estimate *in-context* the vector fields of ordinary differential equations (d'Ascoli et al., 2024), the rate matrices of Markov jump processes (Berghaus et al., 2024), the intensity functions of point processes (Berghaus et al., 2025), and parametric functions for time series imputation (Seifner et al., 2024) and forecasting (Dooley et al., 2024). This emerging paradigm is characterized by a general pretraining strategy, which typically unfolds in three steps. First, one constructs a broad *prior* probability distribution over the space of target functions and, consequently, over the space of dynamical systems they define. This distribution is to represent one's beliefs about the general class of systems one expects to encounter in practice. Second, one samples target functions from this distribution, simulates the corresponding dynamical systems, and corrupts the simulated paths to generate a dataset of noisy observations and target function pairs. This step effectively defines a type of *supervised, meta-learning task* that amortizes the estimation process[2]. Third, one trains a neural network model to match these observation-(target-)function pairs in a supervised way. In this work, we apply this general strategy to the SDE discovery problem (see Figure 1) and make the following contributions:

1. Introduce a data generation model for *sampling and corrupting* SDE paths, restricted to SDEs with polynomial drift and polynomial diffusion. We empirically show that data generated by this model encodes a strong prior, enabling models trained on it to *generalize* to both in-distribution SDEs and real-world datasets.

2. Train the first neural recognition model capable of estimating *in-context* the drift and diffusion functions of *low-dimensional SDEs* from noisy data. We demonstrate that its zero-shot performance matches symbolic, Gaussian process, and neural SDE methods across eight canonical systems, and four real-world datasets, covering processes of one to three dimensions, dataset sizes of $5K$–$20K$ samples, and inter-observation times ranging from $10^{-1}$ to $10^{-5}$ (arbitrary units).

3. Show that the model can be rapidly finetuned to context data, both *dense and sparse*, converging up to $50\times$ faster than neural SDE methods while ultimately outperforming all baselines.

In what follows, we adopt the definition of Bommasani et al. (2021) and name our pretrained model FIM-SDE: Foundation Inference Model for (low-dimensional) SDEs. We briefly revisit related work in Section 2, and formally define the data-driven SDE discovery problem in Section 3. We then present our methodology in Section 4, and demonstrate its capabilities in Section 5. Finally, we discuss limitations and future directions in Section 6.

---

[2]We invite the reader to refer to Appendix A, where we discuss how these ideas relate to other approaches, like the neural process family, or the recently introduced prior fitted networks.

## 2 Related Work

The machine learning community has primarily relied on three approaches to tackle the data-driven SDE discovery problem, namely symbolic regression, Gaussian process (GP) regression, and neural SDEs. Symbolic regression assumes that drift and diffusion can be expressed as linear combinations of basis functions, drawn from a carefully specified dictionary (Boninsegna et al., 2018; Frishman and Ronceray, 2020; Wang et al., 2022). GP regression, by contrast, employs nonparametric priors, modeling the unknown functions as samples from judiciously chosen Gaussian processes (Garcia et al., 2017; Batz et al., 2018). Both approaches suffer from two major limitations. First, they are highly sensitive to the quality and correctness of prior knowledge about the functions they seek to estimate. Second, they assume access to densely sampled data, or else must resort to a variational approximation in order to regularize the inter-observation dynamics (Batz et al., 2018; Duncker et al., 2019; Course and Nair, 2023). While effective in principle, this latter strategy is often undermined by convergence problems (Verma et al., 2024; Adam et al., 2021).

Neural SDE methods (Tzen and Raginsky, 2019; Li et al., 2020; Kidger et al., 2021a; Zeng et al., 2024) offer a prominent deep learning alternative by representing drift and diffusion functions with neural networks, thereby reducing the reliance on prior knowledge. Yet their training involves costly simulations and backpropagation through SDE solvers, typically via adjoint sensitivity methods, which are known to suffer from accuracy issues and slow convergence (Kidger et al., 2021b). More recently, simulation-free Neural SDEs have been proposed (Bartosh et al., 2025), alleviating this computational bottleneck. However, like most traditional machine learning methods, they still require separate optimization for every newly observed system. This constraint severely limits their suitability in real-world scientific workflows, where prior knowledge is scarce and repeated retraining, whether via complex training pipelines or models with slow convergence, quickly becomes impractical.

The goal of this paper is to provide a *zero-shot*, deep learning solution to the data-driven SDE discovery problem that circumvents (most of) these limitations, and thus can readily be deployed in scientific settings. Appendix A provides a more nuanced discussion of related work on the SDE discovery problem.

## 3 Preliminaries

In this section, we concretely define the data-driven SDE discovery problem. First, let us properly introduce the SDE class we will focus on, namely first-order Markov SDEs in the Ito form.

**Ito Stochastic Differential Equations.** A $d$-dimensional SDE in the Ito form is defined as

$$d\mathbf{x} = \mathbf{f}(\mathbf{x})dt + \mathbf{G}(\mathbf{x})d\mathbf{W}(t). \tag{1}$$

The vector-valued function $\mathbf{f} : \mathbb{R}^d \to \mathbb{R}^d$ denotes the state-dependent *drift function* of the process and characterizes the deterministic components of the dynamics. The matrix-valued function $\mathbf{G} : \mathbb{R}^d \to \mathbb{R}^{d \times m}$ denotes the state-dependent *diffusion matrix* and controls the stochastic components which, in turn, are generated through an $m$-dimensional Wiener process $\mathbf{W} : \mathbb{R}^+ \to \mathbb{R}^m$. Given some initial condition $\mathbf{x}(0)$ in $\mathbb{R}^d$, the SDE solution corresponds to a $d$-dimensional stochastic process $\mathbf{x}(t)$, which we call the *state of the system*.

Formally, the drift and diffusion functions are defined as (Gardiner, 2009)

$$f_i(\mathbf{x}) = \lim_{\Delta t \to 0} \frac{1}{\Delta t} \int (x_i' - x_i) p(\mathbf{x}', t + \Delta t | \mathbf{x}, t) d\mathbf{x}', \tag{2}$$

$$[\mathbf{G}(\mathbf{x})\mathbf{G}^T(\mathbf{x})]_{ij} = \lim_{\Delta t \to 0} \frac{1}{\Delta t} \int (x_i' - x_i)(x_j' - x_j) p(\mathbf{x}', t + \Delta t | \mathbf{x}, t) d\mathbf{x}', \tag{3}$$

where $p(\mathbf{x}', t + \Delta t | \mathbf{x}, t)$ denotes the *transition probability* for the state of the system to evolve from $\mathbf{x}$ into $\mathbf{x}'$ under Eq. 1 over the infinitesimal time $\Delta t$. In what follows, we will restrict our attention to diagonal diffusion matrices of the form[3]

$$\mathbf{G}(\mathbf{x}) = \text{diag}(\sqrt{g_1(\mathbf{x})}, \sqrt{g_2(\mathbf{x})}, \ldots, \sqrt{g_d(\mathbf{x})}). \tag{4}$$

---

[3]We opt for diagonal diffusion matrices to limit the number of functions to be inferred. Nothing in the architecture we present below prevents non-diagonal diffusion, and extending pretraining to that setting is straightforward. We leave this extension for future work.

Appendix B provides additional details.

**Data-driven SDE Discovery Problem.** Suppose we are given an ordered sequence of $L$ observations $\mathcal{D}^* = \{(\mathbf{y}_1^*, \tau_1^*), \ldots, (\mathbf{y}_L^*, \tau_L^*)\}$ on some empirical process $\mathbf{y}^*(t) : \mathbb{R}^+ \to \mathbb{R}^d$, recorded at non-equidistant times $0 \leq \tau_1^* < \cdots < \tau_L^* \leq T^*$, with $T^*$ denoting the observation horizon. Now *assume* these observations are generated conditioned on a hidden process $\mathbf{x}^*(t) : \mathbb{R}^+ \to \mathbb{R}^d$, governed by an SDE of the form defined in Eq. 1, with *unknown* drift $\mathbf{f}^*(\mathbf{x})$ and diffusion $\mathbf{G}^*(\mathbf{x})$. Our objective is to construct non-parametric estimators $\hat{\mathbf{f}}_\theta(\mathbf{x}|\mathcal{D}^*)$ and $\hat{\mathbf{G}}_\theta(\mathbf{x}|\mathcal{D}^*)$ — parametrized by $\theta$ and conditioned on the context data $\mathcal{D}^*$ — that best approximate the putative drift $\mathbf{f}^*(\mathbf{x})$ and diffusion $\mathbf{G}^*(\mathbf{x})$ over some "reasonable" domain[4] $\mathcal{X} \in \mathbb{R}^d$. Conventional machine learning approaches invariably address this problem by fitting $\theta$ directly to $\mathcal{D}^*$. As a result, traditional models cannot generally be reused to estimate the SDE of a second empirical process $\mathbf{y}^{**}(t)$, even when the latter is "similar" (*i.e.*, with respect to some metric) to $\mathbf{y}^*(t)$. The methodology we introduce in the next section strongly departs from this tradition.

**Notation.** We use $\mathbf{x}(t)$ to denote simulated SDE paths, and $\mathbf{x}^*(t)$ to denote hidden SDE processes assumed to underlie observed data. Similarly, $\mathbf{y}(t)$ refers to artificially corrupted SDE paths, while $\mathbf{y}^*(t)$ denotes target data. Furthermore, we distinguish between the simulation discretization step $\Delta t$ and the empirical inter-observation times $\Delta \tau$. Finally, we use the same symbols to denote probability distributions and their densities, as well as random variables and their realizations.

## 4 Foundation Inference Models For In-Context Learning

In this section, we propose a methodology for *in-context estimation* of drift and diffusion functions from data, which involves training a neural network model to map a large set of corrupted SDE paths into their *a priori-known* drift and diffusion functions. The latter being sampled from a heuristically constructed synthetic distribution. Thus, the methodology consists of two components: a data generation model and a neural recognition model.

The success of this methodology — namely, that a model (pre)trained solely on synthetic data can help understand and predict unseen empirical processes $\mathcal{D}^*$ — will hinge not only on the inductive biases encoded in the architecture of the recognition model, but also on the family of SDEs represented in the synthetic dataset. Our primary assumption is that, even though we can only consider a very restricted family of drift and diffusion functions, the solution of the SDEs they define can exhibit sufficient complexity to mimic real-world empirical datasets. This assumption resonates with a classical idea popularized by Stephen Wolfram (Wolfram and Gad-el Hak, 2003), but that can be traced back to Kadanoff himself (Kadanoff, 1986, 1987), namely that *simple rules can create complex patterns*. In what follows, we first introduce our data generation model and then present FIM-SDE, our transformer-based, neural-operator recognition model.

### 4.1 Data Generation Model

In order to easily explain the different steps involved in our synthetic data generation process, let us define the probability of observing the noisy sequence $\mathbf{y}_1, \ldots, \mathbf{y}_L \in \mathbb{R}^d$, at the observation times $0 \leq \tau_1 < \cdots < \tau_L \leq T$ for a given time horizon $T$. We write this probability as follows

$$\prod_{i=1}^{L} p_{\text{noise}}(\mathbf{y}_i|\mathbf{x}(\tau_i), \mathbf{f}, \mathbf{G}) p_{\text{FP}}(\mathbf{x}(\tau_i)|\mathbf{f}, \mathbf{G}, \mathbf{x}(\tau_{i-1})) p(\mathbf{x}(0)) p_{\text{grid}}(\tau_1, \ldots, \tau_L) p_{\text{diff}}(\mathbf{G}|d) p_{\text{drift}}(\mathbf{f}|d), \quad (5)$$

where each term represents one step in the data generation process — from the drift and diffusion function generation, to the corruption of the simulations. Let us specify each of them.

**Drift Function Generation**. Drift functions are sampled from the distribution $p_{\text{drift}}(\mathbf{f}(\mathbf{x})|d)$, which is conditioned on the system dimensionality $d$, and is defined to factorize as $p_{\text{drift}}(f_1(\mathbf{x})) \ldots p_{\text{drift}}(f_d(\mathbf{x}))$. Following Berghaus et al. (2024) and d'Ascoli et al. (2024), we generate drift functions for processes of different dimensionalities, ranging from one until some maximum dimension $d_{\text{max}}$. This ensures the applicability of our methodology across systems whose dimensions lie within this range. The set

---

[4]By "reasonable" we loosely mean the subdomain $\mathcal{X} \in \mathbb{R}^d$ that contains the typical set (Thomas and Joy, 2006) of the empirical process $\mathbf{y}^*(t)$.

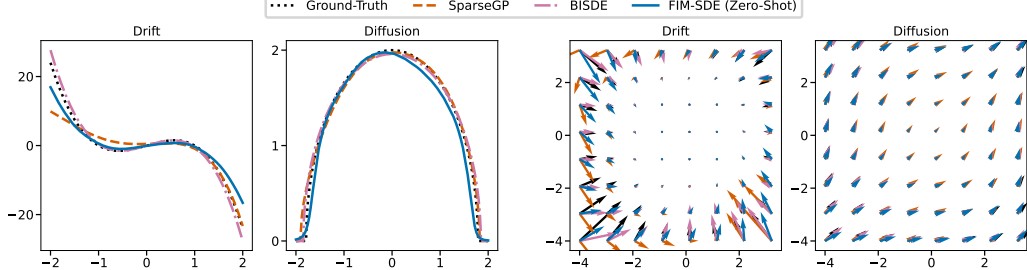

Figure 2: Drift and diffusion estimation for two canonical SDEs with state-dependent diffusion in the uncorrupted setting ($\rho = 0.0$, $\Delta\tau = 0.002$). *Left*: Double-well system (Eq. 37). *Right*: Synthetic 2D system (Eqs. 38 and 39). `FIM-SDE` infers the target functions *in-context* (*i.e.*, zero-shot mode), showing excellent agreement with the ground truth.

of distributions $p_{\text{drift}}(f_i(\mathbf{x}))$, for $i \in 1 \ldots d$, should reflect our beliefs about the class of drift functions likely to arise in nature. In this work, we define them over the space of polynomials of degree at most three, with randomly sampled coefficients. Albeit simple, this choice covers a broad range of commonly used physical models, including chaotic systems. See, for example, the large collection of drift functions in ODEBench (d'Ascoli et al., 2024). We invite the reader to check our discussion about the limitations of this choice in Section 6.

**Diffusion Function Generation**. Diffusion functions are sampled from the distribution $p_{\text{diff}}(\mathbf{G}(\mathbf{x})|d)$, which also factorizes as $p_{\text{diff}}(g_1(\mathbf{x})) \ldots p_{\text{diff}}(g_d(\mathbf{x}))$, where the $g_i(\mathbf{x})$, for $i \in 1 \ldots d$, correspond to the arguments of the square roots in Eq. 4. State-dependent diffusion is common in financial applications and also plays a role in transport problems (see *e.g.* the classical work by Büttiker (1987)). We define $p_{\text{diff}}(g(\mathbf{x}))$ over the space of *positive* polynomials of degree at most two, with random coefficients. This construction covers both geometric Brownian motion and constant diffusion, among others.

**SDE Simulation**. The term $p_{\text{FP}}(\mathbf{x}(\tau_i)|, \mathbf{f}, \mathbf{G}, \mathbf{x}(\tau_{i-1}))$ denotes the transition probability between the states $\mathbf{x}(\tau_{i-1})$ and $\mathbf{x}(\tau_i)$, and corresponds to the solution of the Fokker-Planck equation defined by $\mathbf{f}$ and $\mathbf{G}$ over the interval $[\tau_{i-1}, \tau_i]$. This equation evolves the stochastic process at the "macroscopic" (or path-ensemble) level (Gardiner, 2009). In practice, we define $p(\mathbf{x}(\tau_0 = 0))$ as the standard normal distribution and simulate individual sample paths by numerically solving the corresponding SDE via the Euler-Maruyama scheme, using a discretization step $\Delta t$ until the time horizon $T$.

**SDE Corruption**. Empirical data is noisy, often recorded at irregular time intervals, and can feature inter-observation gaps much larger than the microscopic timescale of the dynamics (*i.e.* $\Delta\tau \gg \Delta t$). The distribution $p_{\text{grid}}(\tau_1, \ldots, \tau_l)$ represents the uncertainty in the sequence of recording times and we implement it by subsampling our SDE solutions through different schemes. Similarly, the distribution $p_{\text{noise}}(\mathbf{y}_i|\mathbf{x}(t), \mathbf{f}, \mathbf{G})$ is defined to represent *additive* Gaussian noise — the maximum entropy attractor of empirical noise distributions (Jaynes, 2003) — with a variance that depends on the characteristic scale of the system's dynamics.

Please refer to Appendix C for the implementation details of all these distributions.

We use the data generation model defined in Eq. 5 to sample a synthetic dataset consisting of tuples of the form $(\mathcal{D}, (\mathbf{f}, \mathbf{G}))$. Here, $\mathcal{D}$ denotes a set of $K$ corrupted time series $\{\mathbf{y}_{k1}, \ldots, \mathbf{y}_{kL}\}_{k=1}^{K}$, where each series contains $L$ observations sampled from the SDE defined by the pair $(\mathbf{f}, \mathbf{G})$. Each tuple represents one distinct SDE, and we generate SDEs of varying dimensionalities.

### 4.2 `FIM-SDE`: a Transformer-based, Neural Operator Recognition Model

We now introduce `FIM-SDE`, a neural recognition model designed to process instances of $\mathcal{D}$, and produce estimates of the target pair $(\mathbf{f}, \mathbf{G})$ over our "reasonable" domain $\mathcal{X}$. To implement this map, we draw on concepts from neural operators, specially DeepONets (Lu et al., 2021). However, different SDEs — and thus, each tuple $(\mathcal{D}, (\mathbf{f}, \mathbf{G}))$ in our dataset — are characterized by distinct spatial and temporal scales. To ensure that `FIM-SDE` is scale-agnostic, we first normalize every $\mathcal{D}$ and renormalize the associated $(\mathbf{f}, \mathbf{G})$ pair accordingly. Appendix D.1 provides the details.

Table 1: MMD evaluation on canonical SDE systems under varying noise levels $\rho$ and inter-observation times $\Delta\tau$. `FIM-SDE` is compared against Symbolic (`BISDE`) and GP (`SparseGP`) regression. Results show the mean and standard deviation over five runs (lower is better). Superscript Roman numerals indicate the number of failed estimations, while the character "$-$" denotes that all runs failed. Bold entries mark the best results. All values are scaled by a factor of 100.

| $\rho$ | $\Delta\tau$ | Model | Double Well | 2D-Synt (Wang) | Damped Linear | Damped Cubic | Duffing | Glycolysis | Hopf |
|---|---|---|---|---|---|---|---|---|---|
| 0.0 | 0.002 | SparseGP | $3(4)^{\text{I}}$ | $\mathbf{0.3(3)}^{\text{I}}$ | $0.9(2)^{\text{I}}$ | $0.5(4)$ | $-$ | $0.68(0)^{\text{IV}}$ | $\mathbf{0.4(3)}$ |
| | | BISDE | $\mathbf{0.2(4)}$ | $\mathbf{0.4(5)}$ | $\mathbf{0.0(0)}$ | $\mathbf{0.03(6)}^{\text{II}}$ | $\mathbf{0.0(0)}$ | $\mathbf{0.3(3)}^{\text{I}}$ | $\mathbf{0.4(4)}$ |
| | | FIM-SDE | $\mathbf{0.5(5)}$ | $\mathbf{0.5(5)}$ | $\mathbf{0.0(0)}$ | $0.9(5)$ | $0.6(5)$ | $1(2)$ | $1(5)$ |
| | 0.02 | SparseGP | $12(2)^{\text{I}}$ | $12(1)$ | $6.67(0)^{\text{IV}}$ | $-$ | $-$ | $-$ | $13.6(5)$ |
| | | BISDE | $\mathbf{0.0(0)}^{\text{I}}$ | $\mathbf{0.1(2)}^{\text{I}}$ | $\mathbf{0.0(0)}$ | $0.7(5)$ | $\mathbf{0.0(0)}$ | $\mathbf{0.02(3)}$ | $\mathbf{0.0(0)}^{\text{I}}$ |
| | | FIM-SDE | $1(1)$ | $\mathbf{0.2(4)}$ | $\mathbf{0.0(0)}$ | $\mathbf{0.1(1)}$ | $\mathbf{0.0(0)}$ | $0.2(2)$ | $0.23(9)$ |
| 0.05 | 0.002 | SparseGP | $5(2)^{\text{II}}$ | $9(2)^{\text{I}}$ | $9(2)^{\text{III}}$ | $18(2)$ | $-$ | $11(2)^{\text{III}}$ | $10(1)$ |
| | | BISDE | $\mathbf{3.1(8)}$ | $9(2)$ | $11.8(2)^{\text{I}}$ | $18(2)^{\text{II}}$ | $11.9(2)^{\text{II}}$ | $14(4)^{\text{II}}$ | $13(2)^{\text{II}}$ |
| | | FIM-SDE | $\mathbf{2.6(6)}$ | $\mathbf{2.2(8)}$ | $\mathbf{2(2)}$ | $\mathbf{6(2)}$ | $\mathbf{1.8(9)}$ | $\mathbf{3(2)}$ | $\mathbf{1.8(9)}$ |
| | 0.02 | SparseGP | $14(2)^{\text{II}}$ | $17(1)$ | $17(8)$ | $-$ | $19.42(0)^{\text{IV}}$ | $-$ | $18(1)$ |
| | | BISDE | $\mathbf{0.3(5)}$ | $\mathbf{0.6(4)}$ | $3(1)$ | $1.7(5)$ | $4(1)$ | $2.1(3)^{\text{II}}$ | $\mathbf{0.4(3)}$ |
| | | FIM-SDE | $1(1)$ | $\mathbf{0.4(6)}$ | $\mathbf{1.1(5)}$ | $\mathbf{0.5(3)}$ | $\mathbf{1.4(8)}$ | $\mathbf{0.8(2)}$ | $\mathbf{0.2(2)}$ |

A direct inspection of Eqs. 2 and 3 reveals that the values of the drift and diffusion functions *at a given location*, say $\mathbf{x}$, only depend on transitions that take place in the "neighborhood" of $\mathbf{x}$. In other words, the sequential nature of the time series in $\mathcal{D}$ does not provide additional information for estimating the local expectations. We therefore reorganize each set $\mathcal{D}$ into a collection of $K(L-1)$ transition tuples $\tilde{\mathcal{D}} = \{\mathbf{y}_i, \Delta\mathbf{y}_i, \Delta\mathbf{y}_i^2, \Delta\tau_i\}_{i=1}^{K(L-1)}$, where each tuple only keeps information about one-step transitions[5], and where we explicitly include both $\Delta\mathbf{y}_i, \Delta\mathbf{y}_i^2$ to enable the network we define below to learn separate feature channels for drift and diffusion estimation. Intuitively, a model that estimates $\mathbf{f}$ or $\mathbf{G}$ at location $\mathbf{x}$ from $\tilde{\mathcal{D}}$, should dynamically *query* every transition in $\tilde{\mathcal{D}}$, and weight them according to their spatial location (*i.e.* the $\mathbf{y}_i$) with respect to $\mathbf{x}$, to compute the relevant local expectation. Thus, attention mechanisms seem like the natural choice.

We embed all transitions in $\tilde{\mathcal{D}}$ using a *self-attentive encoder*, and introduce a *functional attention mechanism* that performs location-dependent querying over the set of embedded transitions. Let $\phi^\theta(\cdot)$ and $\Phi^\theta(\cdot)$ denote linear projections and feed-forward neural networks, respectively. Let $\psi^\theta(\cdot, \cdot, \cdot)$ denote attention layers, and let $\Psi^\theta(\cdot, \cdot, \cdot)$ denote Transformer encoders with linear attention (Katharopoulos et al., 2020; Shen et al., 2021), both of which take three arguments as inputs (*i.e.* queries, keys and values). Finally, let $\theta$ denote model parameters.

**Context Matrix** (DeepONet's *Branch-net* equivalent). Let us define the $n$-dimensional embeddings

$$\mathbf{d}_i^\theta = \text{concat}\left[\phi_{\mathbf{y}}^\theta(\mathbf{y}_i), \phi_{\Delta\mathbf{y}}^\theta(\Delta\mathbf{y}_i), \phi_{\Delta\mathbf{y}^2}^\theta(\Delta\mathbf{y}_i^2), \phi_{\Delta\tau}^\theta(\Delta\tau_i)\right], \tag{6}$$

where $i$ runs from 1 to $K(L-1)$ for every element in $\tilde{\mathcal{D}}$, and the linear projections $\phi^\theta$ map their inputs onto $\mathbb{R}^{n/4}$. Let now $\mathbf{D}^\theta = [\mathbf{d}_1^\theta, \ldots, \mathbf{d}_{K(L-1)}^\theta]$ denote the $n \times K(L-1)$ matrix of linear embeddings. We define the (self-attentive) *context matrix*

$$\tilde{\mathbf{D}}^\theta = \Psi^\theta(\mathbf{D}^\theta, \mathbf{D}^\theta, \mathbf{D}^\theta), \text{ so that } \tilde{\mathbf{D}}^\theta \in \mathbb{R}^{n \times K(L-1)}, \tag{7}$$

which encodes the entire *context* data $\tilde{\mathcal{D}}$.

**Functional Attention Mechanism** (DeepONet's *Trunk-net* equivalent). Let us assume we want to estimate the drift function at location $\mathbf{x}$. Let $\mathbf{x}$ be the input query to a sequence of $M$ attention networks $\psi_{\mathbf{f},i}^\theta$, and let $\tilde{\mathbf{D}}^\theta$ be both keys and values to every attention network in that sequence. We define a set of $M + 1$ location-dependent embeddings for the estimator of $\mathbf{f}$ as

$$\mathbf{h}_i^\theta(\mathbf{x}|\tilde{\mathcal{D}}) = \psi_{\mathbf{f},i}^\theta(\mathbf{h}_{i-1}^\theta(\mathbf{x}|\tilde{\mathcal{D}}), \tilde{\mathbf{D}}^\theta, \tilde{\mathbf{D}}^\theta), \tag{8}$$

---

[5]That said, note that the $\Delta\tau$ in $\tilde{\mathcal{D}}$ are typically much larger than the $\Delta t$ characteristic of our simulations.

Table 2: MMD evaluation on data sampled from the Lorenz attractor under three initial conditions: $\mathcal{N}(0,1)$, $(1,1,1)$ and $\mathcal{N}(0,2)$. `LatentSDE` was trained for 5000 iterations on data from $\mathcal{N}(0,1)$, while `FIM-SDE`, by contrast, required only 100 finetuning iterations on the same dataset.

| Model | $\mathcal{N}(0,1)$ | $(1,1,1)$ | $\mathcal{N}(0,2)$ |
|---|---|---|---|
| `LatentSDE` | 0.051708 | 0.135003 | 0.054335 |
| `FIM-SDE` (Zero-Shot) | 0.403933 | 1.151080 | 0.378515 |
| `FIM-SDE` (Finetuned) | **0.003838** | **0.037159** | **0.003335** |

where $i$ runs from 1 until $M$, and the first embedding is defined as $\mathbf{h}_0^\theta(\mathbf{x}|\tilde{\mathcal{D}}) := \mathbf{h}_0^\theta(\mathbf{x}) = \phi_\mathbf{x}^\theta(\mathbf{x})$. We then compute $\hat{\mathbf{f}}_\theta(\mathbf{x}|\tilde{\mathcal{D}}) = \Phi_\mathbf{f}^\theta(\mathbf{h}_M^\theta(\mathbf{x}|\tilde{\mathcal{D}}))$. The calculation of $\hat{\mathbf{G}}_\theta(\mathbf{x}|\tilde{\mathcal{D}})$ is analogous, and Figure 1 illustrates the `FIM-SDE` architecture.

**Target Objective**. We adopt the mean-squared error as target objective, and define

$$\mathcal{L}_1^\theta(\mathbf{x}, \tilde{\mathcal{D}}, \mathbf{f}, \mathbf{G}) = \sum_{i=1}^d \left\{ \left( \hat{f}_{i,\theta}(\mathbf{x}|\tilde{\mathcal{D}}) - f_i(\mathbf{x}) \right)^2 + \left( \sqrt{\hat{g}_{i,\theta}(\mathbf{x}|\tilde{\mathcal{D}})} - \sqrt{g_i(\mathbf{x})} \right)^2 \right\}, \qquad (9)$$

which can be computed by marginalizing $\tilde{\mathcal{D}}$, $\mathbf{f}$ and $\mathbf{G}$ under the generative model (Eq. 5), and $\mathbf{x}$ under *e.g.* a uniform distribution over $\mathcal{X}$. Now, one can readily show that this objective reaches a global minimum when the estimators $\hat{\mathbf{f}}_\theta$ and $\hat{\mathbf{G}}_\theta$ respectively match the mean of the true posterior distributions $p_{\text{drift}}(\mathbf{f}|\tilde{\mathcal{D}})$ and $p_{\text{diff}}(\mathbf{G}|\tilde{\mathcal{D}})$. However, the reader will immediately notice that marginalizing $\mathcal{L}_1^\theta$ under a uniform distribution can result in numerical instabilities, caused by some local contributions dominating the integral, not due to the objective itself, but because of the norm of the involved functions in those regions. We therefore need a weighting mechanism to balance these contributions over $\mathcal{X}$. Consider the loss

$$\mathcal{L}_\theta = \mathop{\mathbb{E}}_{\mathbf{x} \sim \mathcal{U}(\mathcal{X})} \mathop{\mathbb{E}}_{(\tilde{\mathcal{D}},(\mathbf{f},\mathbf{G})) \sim p_{\text{prior}}} \left[ e^{-U_\theta(\mathbf{x},\tilde{\mathcal{D}})} \mathcal{L}_1^\theta(\mathbf{x}, \tilde{\mathcal{D}}, \mathbf{f}, \mathbf{G}) + U_\theta(\mathbf{x}, \tilde{\mathcal{D}}) \right], \qquad (10)$$

where $\mathcal{U}$ denotes uniform distribution, and $p_{\text{prior}}$ labels our generative model (Eq. 5). The learnable function $U_\theta(\mathbf{x}, \tilde{\mathcal{D}})$ provides adaptive weights, and can be interpreted as an epistemic uncertainty estimator (see *e.g.* Appendix B in Karras et al. (2024)). Indeed, assuming that $\mathcal{L}_1^\theta$ is a location-dependent Gaussian variable, $U_\theta$ acts as its log-variance. From Eq. 10, we see that large $U_\theta$ values down-weight regions where the model is uncertain, while the additive $+U_\theta$ term prevents it from collapsing to $+\infty$. Empirically, we have observed that $U_\theta$ peaks in sparsely observed or highly noisy areas, acting as a proxy for epistemic uncertainty. We implement $U_\theta$ with a third set of $M$ attention functions $(\psi_{U,1}^\theta, \ldots, \psi_{U,M}^\theta)$, but only back-propagate up to $\tilde{\mathbf{D}}^\theta$ (*i.e.* we detach $U_\theta$).

**Finetuning**. `FIM-SDE` supports rapid finetuning *on both* densely and sparsely sampled data[6]. For densely sampled data, the model is finetuned by maximizing the likelihood of short-time transitions induced by the estimators $\hat{\mathbf{f}}_\theta$, $\hat{\mathbf{G}}_\theta$ with respect to the target data. For sparsely sampled data, finetuning proceeds instead by simulating the inferred SDE (defined by $\hat{\mathbf{f}}_\theta$, $\hat{\mathbf{G}}_\theta$) for $\ell$ steps between observation pairs, and minimizing the discrepancy between the last simulation step and its target value. Appendix D.4 gives further details and the explicit loss functions used for finetuning.

Let us close this section by noting that, unlike traditional approaches, both neural and not, that rely on variational methods to regularize the dynamics between sparse observations, `FIM-SDE` requires no such explicit regularization. *Its pretraining implicitly enforces it.* We demonstrate this empirically below.

## 5 Experiments

What follows outlines the experimental setup, including pretraining details, datasets, evaluation metrics, and baselines.

**Pretraining.** We (pre)trained a single 20M-parameter instance of `FIM-SDE` on a synthetic dataset of 600K *low-dimensional SDEs*. The dataset contains three-, two-, and one-dimensional systems in

---

[6]The target data may consist of context data $\tilde{\mathcal{D}}$, or additional samples from the same underlying process $\mathbf{y}^*$.

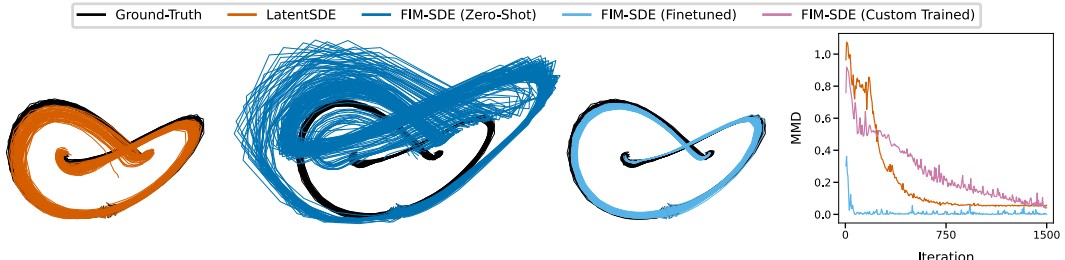

Figure 3: Sample paths (first three panels) and MMD convergence (right) for the Lorenz experiment. `FIM-SDE` in zero-shot mode already captures the global dynamics, though errors accumulate over time. With finetuning, `FIM-SDE` quickly refines its estimates of the dynamics, requiring only a few iterations and converging much faster than `LatentSDE`.

a 3:2:1 ratio, with context sizes (*i.e.*, $|\tilde{\mathcal{D}}|$) ranging from 128 to 12800 tuples, and inter-observation times (*i.e.*, $\Delta\tau$) spanning $10^{-1}$ to $10^{-3}$ (arbitrary units). Further details on the (pre)training data, architecture and hyperparameters, as well as ablation studies, are provided in Appendix C, D, and H, respectively.

**Datasets.** We evaluate `FIM-SDE` on two classes of systems: eight canonical SDEs of varying dimensionality and complexity, and four real-world systems derived from experimental data. The canonical systems are defined in Appendices G.1 and G.2, and belong to the SDE class used to construct the prior distribution (Eq. 5). That is, SDEs with polynomial drift of dimension at most three, and polynomial diffusion of dimension at most two. We denote this class by $\mathcal{C}$ in what follows. The experimental data consist of stock prices of Facebook and Tesla, oil price fluctuations, and wind speed recordings (Wang et al., 2022).

**Baselines.** We compare against three baselines: the GP regression model of Batz et al. (2018), the symbolic regression model of Wang et al. (2022), and the neural SDE model of Li et al. (2020). Let us referred to them as `SparseGP`, `BISDE`, and `LatentSDE`, respectively. Appendix E contains details about their implementation, hyperparameters and training.

**Metrics.** The quality of the inferred SDEs is assessed using three complementary metrics. First, we compute the maximum mean discrepancy (MMD) between simulated and held-out path ensembles, using the signature kernel of Király and Oberhauser (2019). Second, when ground-truth functions are available, we measure the mean-squared error (MSE) between estimated and true drift/diffusion functions on a predefined grid. And third, we qualitatively assess the results by inspecting the inferred functions on uniform grids and comparing sampled paths. Implementation details for both MMD and MSE are provided in Appendix F.

## 5.1 Canonical SDE systems

**Comparison against Symbolic and GP regression**. We begin by studying seven canonical SDEs, defined in Appendix G.1, spanning diverse non-linear behavior such as double-well dynamics and Hopf bifurcations. Each system is simulated with an Euler-Maruyama step of $\Delta t = 0.002$, which fixes the "microscopic" time scale of the dynamics. To assess robustness of `FIM-SDE` and the baselines under different forms of corruption, we design four experimental settings per target SDE. Specifically, we introduce additive noise with variance $\rho$ and vary the inter-observation time. The first setting preserves the microscopic scale, $\Delta\tau = \Delta t = 0.002$, while the second uses a coarser resolution, $\Delta\tau = 0.02$, but keeps the number of observations per path fixed. Both baselines assume (by construction) access to the clean system states, and are therefore expected to perform well on dense data (*i.e.*, $\Delta\tau \simeq \Delta t$) without noise corruption. In contrast, `FIM-SDE` is (pre)trained on corrupted paths characterized by a broad inter-observation time distribution $p(\Delta\tau)$ (see Appendix C.5). We use `FIM-SDE` and the baselines to estimate the drift and diffusion functions that best characterize the sampled paths, repeating each experiment five times with independently generated datasets. Both baselines must be retrained from scratch for every system and configuration, whereas `FIM-SDE` is applied off-the-shelf, without modification, across all experiments.

Table 3: MMD in four real-world systems. Per system, we report the mean and standard deviation across five cross-validation splits. `LatentSDE` has been trained for 5000 iterations, while `FIM-SDE` has been finetuned for only 100 iterations. `FIM-SDE` (Finetuned) achieves the overall best performance across the different systems.

| Model | Wind | Oil | Facebook | Tesla |
|---|---|---|---|---|
| BISDE | $\mathbf{0.000 \pm 0.000}$ | $0.137 \pm 0.097$ | $0.053 \pm 0.060$ | $0.013 \pm 0.015$ |
| LatentSDE | $0.350 \pm 0.212$ | $0.235 \pm 0.167$ | $0.014 \pm 0.018$ | $0.016 \pm 0.023$ |
| FIM-SDE (Zero-Shot) | $0.330 \pm 0.114$ | $0.199 \pm 0.137$ | $0.012 \pm 0.026$ | $\mathbf{0.000 \pm 0.000}$ |
| FIM-SDE (Finetuned) | $0.029 \pm 0.034$ | $\mathbf{0.068 \pm 0.042}$ | $\mathbf{0.006 \pm 0.007}$ | $0.004 \pm 0.008$ |

Table 1 reports the MMD evaluations across all four experimental setting, and all seven SDE systems, averaged over five runs. The emergent picture is clear: `FIM-SDE` consistently outperforms both baselines on systems corrupted by noise. On clean systems, `BISDE` tends to perform best, although `FIM-SDE` attains comparable MMD values in roughly half of the cases. Surprisingly, both `BISDE` and `SparseGP` frequently produce estimates that correspond to invalid SDEs (even in the uncorrupted settings!). `FIM-SDE` is stable in this regard. Table 6 (Appendix) reports MSEs between estimated and target functions, providing a consistent and complementary picture. Moving on to a qualitative view, Figure 2 shows the inferred drift and diffusion for the double-well system (Eq. 37), and a synthetic 2D system (Eqs. 38, 39), in the uncorrupted setting. `FIM-SDE` estimations are in excellent agreement with the ground-truth. Additional inferred functions and sampled paths for the remaining systems are shown in Figure 5 (Appendix).

The results presented here show that `FIM-SDE` successfully *generalizes to unseen SDEs within* $\mathcal{C}$ (the SDE class in our prior distribution). Appendix H provides ablation studies on the effects of varying the parameter count, the definition of $\mathcal{C}$, and the context size of the model on these canonical systems.

**Comparison against Neural SDE methods**. We now turn to the celebrated Lorenz attractor, weakly perturbed by constant diffusion. The system is simulated with initial conditions sampled from $\mathcal{N}(0, 1)$, observed *sparsely* in time with gaps of $\Delta\tau = 0.025$, and perturbed by additive Gaussian noise with variance $\rho = 0.01$. This setup reproduces the experimental conditions studied by Li et al. (2020) (see Appendix G.2 for details). We use this data to train `LatentSDE`, which handles data sparsity by modeling the dynamics in latent space, using an inhomogeneous SDE posterior together with a homogeneous SDE prior. The latter is what enables `LatentSDE` to tackle the SDE discovery problem. `FIM-SDE`, in contrast, treats the same data as context, and is evaluated both in *zero-shot* mode and after *finetuning* on its context (using the finetuning cost in Eq. 35 for sparse data).

For evaluation, we first generate dense ground-truth trajectories under three initial conditions, namely $\mathcal{N}(0, 1)$, $(1, 1, 1)$ and $\mathcal{N}(0, 2)$. Then, we compute the MMD between these and samples drawn from the trained `LatentSDE` prior, as well as from `FIM-SDE` and its finetuned counterpart. Table 2 summarizes the results: `LatentSDE` outperforms `FIM-SDE` in the zero-shot setting, but after less than 100 finetuning iterations, `FIM-SDE` surpasses `LatentSDE` on all three tests. We attribute the weak zero-shot performance of `FIM-SDE` to the fact that, despite being in $\mathcal{C}$, the perturbed Lorenz attractor and its samples are drift-dominated (Figure 3 makes this feature apparent). Because our pre-training set contains mostly non-zero diffusion functions, `FIM-SDE` is biased toward inferring finite diffusion, which increases the variance of its generated paths leading to higher MMD scores. Finetuning on the context data quickly removes the bias, yielding accurate estimates. The right panel of Figure 3 further shows that fine-tuning `FIM-SDE` converges more than $5\times$ faster than `LatentSDE`, and over $15\times$ faster than training `FIM-SDE` from scratch, *underscoring the strong initialization provided by the pretrained parameters*.

## 5.2 Real-World Systems

In this section, we study how well `FIM-SDE` generalizes to four real-world stochastic systems obtained from experimental data. Specifically, we consider stock prices of Facebook and Tesla, as well as wind speed and oil price fluctuations, originally collected and analyzed by Wang et al. (2022). As shown in Figures 4 and 6 (Appendix), these datasets exhibit inherent stochasticity and are *densely recorded*.

We partition each dataset into five folds, train `BISDE` and `LatentSDE` on four of these, and compute the MMD on the held-out fold, repeating this process across all folds and averaging the results (see Appendix G.3 for details). Within the same cross-validation protocol, `FIM-SDE` uses the data as

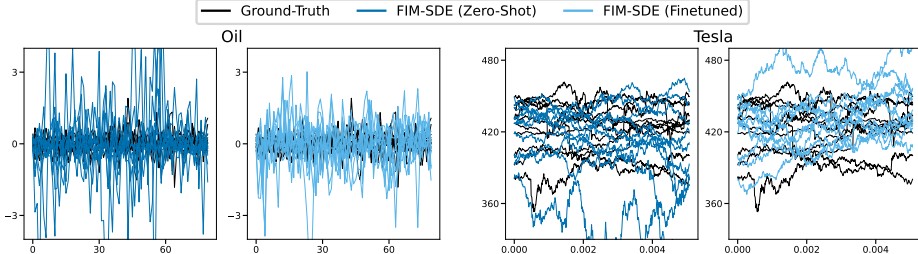

Figure 4: Paths sampled from SDEs inferred by `FIM-SDE` (dark blue) and by `FIM-SDE` finetuned on oil (left) and stock price data (right). Finetuning noticeably improves sample path quality in the oil dataset, while the strong zero-shot performance on stock prices is retained.

context, and is evaluated both in *zero-shot* mode and after finetuning with the dense-data objective (Eq. 34). Table 3 reports the resulting MMD scores.

The zero-shot performance of `FIM-SDE` is competitive overall, and on the stock datasets it even outperforms both `BISDE` and `LatentSDE`. On the wind and oil datasets, zero-shot `FIM-SDE` lags behind `BISDE`, but finetuning substantially improves its estimates, yielding the best performance across all systems. In particular, finetuning adjusts the inferred diffusion in the oil and wind datasets, as illustrated in Figures 4 and 6 (Appendix). This shows that pretraining on our synthetic, prior distribution over the SDE class $\mathcal{C}$ (Eq. 5) encodes useful inductive biases into the model, while finetuning adapts it to dataset-specific features.

Finally, Table 9 (Appendix) compares the training trajectories of `LatentSDE` and `FIM-SDE`. Whereas `LatentSDE` requires thousands of iterations to converge, `FIM-SDE` surpasses it after only 100 finetuning iterations, *representing a $50\times$ reduction in compute.* Taken together, these findings demonstrate that `FIM-SDE` functions as a true foundation model: it achieves competitive zero-shot performance and can be rapidly adapted to complex real-world data through finetuning.

## 6 Conclusions

In this work, we introduced a methodology for *in-context* learning of stochastic differential equations (SDEs), built on two components: (i) a prior distribution $p_{\text{prior}}(\mathcal{C})$ over the SDE class $\mathcal{C}$, consisting of polynomial drift of degree $\leq 3$ and polynomial diffusion of degree $\leq 2$; and (ii) `FIM-SDE`, a neural recognition model pretrained on $p_{\text{prior}}(\mathcal{C})$. We empirically showed that the *zero-shot* performance of `FIM-SDE` is on par with symbolic, Gaussian process, and neural SDE methods across eight canonical systems and four real-world datasets. Moreover, we showed that `FIM-SDE` can be *rapidly finetuned* on both dense and sparse data, consistently outperforming all baselines after only a few iterations.

*The main limitation of our methodology* lies in the prior distribution, which is currently restricted to (i) the SDE class $\mathcal{C}$ of low-degree polynomial drift and diffusion, and (ii) low-dimensional systems. The practical impact of the first restriction depends on whether $\mathcal{C}$ is rich enough to capture SDEs relevant in real-world applications. Our experiments indicate that $\mathcal{C}$ offers not only useful inductive biases but also a strong initialization for finetuning. Future work will explore more expressive priors, such as random function compositions via unary-binary trees (Lample and Charton, 2020), to generate more complex drift functions.

The restriction to low-dimensional systems stems mainly from the rejection rates in data generation, which increase sharply with dimension: 50% in 1D, 78% in 2D, and 92% in 3D (curse of dimensionality). In future work, we aim to reduce rejection rates by generating SDEs known a priori to admit solutions, e.g. by randomly sampling globally Lipschitz drift and diffusion functions with linear growth (Øksendal, 2010). This would enable the simulation of higher-dimensional SDEs. Still, many complex and relevant dynamical systems are, in fact, low dimensional.

Finally, future work will also explore leveraging the pretrained `FIM-SDE` weights to regularize equation discovery from high-dimensional data, with applications ranging from neural population and chemical reaction dynamics (Duncker et al., 2019) to the evolution of natural language content (Cvejoski et al., 2023, 2022).

## Acknowledgments and Disclosure of Funding

This research has been funded by the Federal Ministry of Education and Research of Germany and the state of North-Rhine Westphalia as part of the Lamarr Institute for Machine Learning and Artificial Intelligence.

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

# A Related Work

**Variational Inference and Parametric Priors**. In general, when the observed data is noisy, sparse in time, or both, one faces uncertainty not only in determining the drift and diffusion of the putative SDE, but also in the state of the system itself. Therefore, a Bayesian treatment requires the estimation of the posterior distribution over these states, conditioned on the noisy data (*i.e.* the so-called smoothing problem). Starting with the seminal works of Archambeau et al. (2007a,b), which approximated the posterior over the states with a variational and inhomogeneous Gaussian process, most proposals have mainly focused on devising different strategies to infer the smoothing distribution — *while assuming prior parametric forms for the drift and diffusion functions*. See *e.g.* Vrettas et al. (2011), Wildner and Koeppl (2021), or the recent work by Verma et al. (2024).

**Non-parametric Priors**. A notable exception is the proposal of Duncker et al. (2019), which extended the variational trick of Archambeau et al. (2007a) by imposing a non-parametric prior over the drift of the prior process. However, Archambeau et al. (2007a)'s trick has been shown to suffer from significant convergence issues (Verma et al., 2024), which are inherited by Duncker et al. (2019)'s model. In contrast, Batz et al. (2018) framed the drift-diffusion estimation problem from uncorrupted (*i.e.* clean and dense) data as a Gaussian process regression problem, and extended this (non-parametric) approach to observations that are sparse in time, by leveraging Orstein-Uhlenbeck bridges optimized with an expectation maximization algorithm. Nevertheless, this extension can only deal with non-parametric drifts (*i.e.* the diffusion is restricted to parametric forms) and clean data, and is highly sensitive to the choice of prior hyperparameters.

**Symbolic Regression.** Symbolic regression methods for drift and diffusion estimation mainly extend the SINDy algorithm (Brunton et al., 2016) — which performs sparse linear regression on a predefined library of candidate nonlinear functions — to SDEs. The first of these extensions corresponds to the work of Boninsegna et al. (2018), which sets the regression problem by approximating the local values of the drift and diffusion functions with the empirical expectations of Eqs 2 and 3 of Section 3. However, the calculation of these local expectations generally requires significant amounts of data, even for one-dimensional systems. To (somewhat) alleviate this issue, Huang et al. (2022) and Wang et al. (2022) recently resorted to sparse Bayesian learning, but their solutions still require sizable dataset sizes and, most problematically, assume access to clean and dense observations. Other key contributions came from the physics community. For example, Frishman and Ronceray (2020) leveraged symbolic regression too, but utilized the connection between the Ito and Stratonovich formulation of SDEs to devise an unbiased estimator for the drift and diffusion functions under noisy observations. They were also able to use their methodology to estimate non-equilibrium observables, like the stochastic entropy production. Nevertheless, their empirical results still showed significant sensitivity to noise. In response to these limitations, Course and Nair (2023) proposed a hybrid solution that leveraged the variational trick of Archambeau et al. (2007a), while allowing the drift of the prior process to be approximated by a sparse, linear combination of known basis functions. However, their model inherits the slow convergence problems of the variational approximation. What is more, all symbolic regression methods are limited by construction to linear combination of the functions in their library, which therefore makes their performance highly dependent on the preselected functions within the library. Besides these works, other recent efforts tackle underdamped stochastic systems (Brückner et al., 2020) and stochastic networked (that is, interacting) systems (Gao et al., 2024), and we refer the reader to them for details.

**Neural SDEs.** Most deep learning approaches for SDE learning rely mainly on neural network parameterizations of the drift and diffusion functions — the so-called Neural SDE models — and primarily focus on path generation (that is, smoothing or forecasting tasks). Prominent examples include the works by Tzen and Raginsky (2019), Liu et al. (2019), Li et al. (2020), Kidger et al. (2021a), Biloš et al. (2023) and Zeng et al. (2024). The training of these models is however known to be a formidable hurdle, which many have attempted to overcome (Kidger et al., 2021b). We nevertheless note the very recent work by Bartosh et al. (2025).

**Early Amortization Inference**. Our methodology can be understood as an *amortization* of the probabilistic inference process (of the drift and diffusion functions) through a single neural network, and is therefore akin to the works of Stuhlmüller et al. (2013), Heess et al. (2013) and Paige and Wood (2016). Rather than treating, as these previous works do, our (pre)trained models as auxiliary to Monte Carlo or expectation propagation methods, we employ them to directly estimate the drift and diffusion functions from various synthetic and experimental datasets.

**Conditional Neural Network Models and Meta-Learning**. Different from "foundation models" trained on synthetic datasets (like `FIM-SDE` or Prior Fitted Networks discussed below), conditional neural network models — such as the neural statistician Edwards and Storkey (2016); Hewitt et al. (2018) or members of the neural process family Garnelo et al. (2018b,a); Kim et al. (2019) — are trained across sets of different, albeit related datasets, *each assumed to share a common context latent variable*. These meta-learning models are trained exclusively on (sets of) datasets *from their target domains*, rendering both their optimized weights and the *representations they can infer* (that is, their latent variables) problem/data specific. In contrast, our method maintains the same network parameters and representation semantics throughout all experiments. The representations consistently correspond to drift and diffusion functions of SDEs, regardless of the target dataset.

**Prior-Fitted Networks**. A recent line of work introduces Prior-Fitted Networks (PFNs) (Müller et al., 2022; Hollmann et al., 2022; Müller et al., 2025) as a powerful framework for amortized Bayesian inference. PFNs are typically implemented as transformers trained on large collections of synthetic, supervised-learning tasks sampled from a user-specified prior. Once (pre)trained, these models approximate *posterior predictive distributions* in a single forward pass, effectively mimicking Gaussian processes and other Bayesian inference procedures at test time. Our foundation inference models (FIMs) (Berghaus et al., 2024; Seifner et al., 2024) share the philosophy of pretraining on synthetic data, but target a fundamentally different goal. Rather than learning posterior predictive distributions for supervised tasks, FIMs are designed to infer dynamical systems (*i.e.* mathematical models) from data. In other words, FIMs *infer latent structure underlying target dataset*. Let us close this section by noting that, very recently, the problem of amortized *in-context* Bayesian posterior estimation was framed very precisely and studied by Mittal et al. (2025b,a).

# B    Background: Stochastic Differential Equations

In this section we provide some background information for stochastic differential equations, that we already touched on in Section 3. For a thorough discussion, we refer the reader to (Gardiner, 2009) and (Øksendal, 2010). We recall the definition of stochastic differential equations, detail a path sampling scheme and specify the kind of equations we consider in this work.

## B.1    Definition

A $d$-dimensional stochastic process $\mathbf{x}(t)$ follows an Ito stochastic differential equation (SDE), if it satisfies

$$x_i(\bar{t}) = x_i(\underline{t}) + \int_{\underline{t}}^{\bar{t}} f_i(\mathbf{x}(t'), t') dt' + \sum_{j=1}^{m} \int_{\underline{t}}^{\bar{t}} G_{ij}(\mathbf{x}(t'), t') dW_j(t') \tag{11}$$

for all $i \leq d, \underline{t} < \bar{t}$ and some vector-valued *drift function* $\mathbf{f} : \mathbb{R}^d \times \mathbb{R}^+ \to \mathbb{R}^d$ and *diffusion function* $\mathbf{G} : \mathbb{R}^d \times \mathbb{R}^+ \to \mathbb{R}^{d \times m}$, where $\mathbf{W} : \mathbb{R}^+ \to \mathbb{R}^m$ is a standard $m$-dimensional Wiener process. Such process is denoted as a differential equation

$$d\mathbf{x}(t) = \mathbf{f}(\mathbf{x}(t), t) dt + \mathbf{G}(\mathbf{x}(t), t) d\mathbf{W}(t). \tag{12}$$

Equation (11) contains a deterministic Riemann integral and stochastic Ito integrals (which are defined as Riemann-Stieltjes integrals). As such, (11) is understood as the infinitesimal limit in a discretization of the interval $[\underline{t}, \bar{t}]$ into multiple short-time steps. Considering a short-time step $[t, t + \Delta t]$, the right-hand side of (11) is approximately

$$x_i(t) + f_i(\mathbf{x}(t), t)\Delta t + \sum_{j=1}^{m} G_{ij}(\mathbf{x}(t), t)(W_j(t + \Delta t) - W_j(t)), \tag{13}$$

by the definition of the integrals. The infinitesimal limit is well defined if drift and diffusion are globally Lipschitz and feature (at most) quadratic growth (Øksendal, 2010).

A stochastic process following an Ito SDE is Markovian, emitting a transition probability distribution $p(\cdot, \cdot \mid \mathbf{x}, t)$. Its relationship to the drift and diffusion functions is characterized by two equations:

$$f_i(\mathbf{x}, t) = \lim_{\Delta t \to 0} \frac{1}{\Delta t} \int (x_i' - x_i) p(\mathbf{x}', t + \Delta t \mid \mathbf{x}, t) d\mathbf{x}' \tag{14}$$

$$[\mathbf{G}(\mathbf{x},t)\mathbf{G}(\mathbf{x},t)^T]_{ij} = \lim_{\Delta t \to 0} \frac{1}{\Delta t} \int (x'_i - x_i)(x'_j - x_j)p(\mathbf{x}',t+\Delta t|\mathbf{x},t)d\mathbf{x}' \tag{15}$$

Hence, drift and diffusion can be reinterpreted by infinitesimal limits of mean and covariance of the short-time transitions.

## B.2 Simulating SDEs: Euler-Maruyama Scheme

The formal solution construction via discretization and short-time step approximation (13) is also a scheme to simulate paths (i.e. realizations) of the stochastic process. Consider again a short-time step $[t, t+\Delta t]$. Then given $\mathbf{x}(t)$, the solution at $t+\Delta t$ is approximately

$$\mathbf{x}(t+\Delta t) \approx \mathbf{x}(t) + \mathbf{f}(\mathbf{x}(t),t)\Delta t + \mathbf{G}(x(t),t)(\mathbf{W}(t+\Delta t) - \mathbf{W}(t)) \tag{16}$$

because of (13). By definition of the Wiener process, its increment follows a the multivariate normal distribution

$$(\mathbf{W}(t+\Delta t) - \mathbf{W}(t)) \sim \mathcal{N}(\mathbf{0}, \Delta t\mathbf{I}_m) \tag{17}$$

which enables the sampling of $\mathbf{x}(t+\Delta t) \mid \mathbf{x}(t)$ in (16). In other words, the short-time transition probability is approximately Gaussian:

$$p(\cdot, t+\Delta t \mid \mathbf{x}, t) \approx \mathcal{N}\left(\mathbf{x} + \mathbf{f}(\mathbf{x},t)\Delta t, \, \mathbf{G}(\mathbf{x},t)\mathbf{G}(\mathbf{x},t)^T\Delta t\right) \tag{18}$$

To conclude, the so called *Euler-Maruyama scheme* for sampling a SDE $\mathbf{x}(t)$ on an interval, say $[0, T]$, from an initial state $\mathbf{x}(0)$ involves discretization of the interval into $M$ small time-steps of size $\Delta t = \frac{T}{M}$ and iteratively sampling from (18). Finer discretizations, i.e. smaller $\Delta t$, yield more accurate sample paths.

## B.3 Our Setup, the Shot-time Transition Probability and its Logarithm

Our work focuses on the inference of SDEs with *purely state-dependent* drift and diffusion functions $\mathbf{f} : \mathbb{R}^d \to \mathbb{R}^d$ and $\mathbf{G} : \mathbb{R}^d \to \mathbb{R}^{d \times m}$. We denote these differential equations by

$$d\mathbf{x} = \mathbf{f}(\mathbf{x})dt + \mathbf{G}(\mathbf{x})d\mathbf{W}(t). \tag{19}$$

Moreover, our train data distribution and inference model assumes *diagonal diffusion*. Diffusion functions of this form are uniquely defined by

$$\mathbf{G}(\mathbf{x}) = \mathrm{diag}(\sqrt{g_1(\mathbf{x})}\sqrt{g_2(\mathbf{x})}, \ldots, \sqrt{g_d(\mathbf{x})}), \tag{20}$$

with component functions $g_i : \mathbb{R}^d \to \mathbb{R}^+$ for $i \leq d$.

This diffusion function modeling assumption is not as restrictive it might seem. All baseline models (see Appendix E) make the same assumption. In principle, our framework can handle non-diagonal diffusion. We make this simplifying assumption for a fairer comparison to the baselines and lower computational load during data generation and model training.

Under these assumptions, the approximation of the short-time transition probability from (18) reduces to

$$p(\cdot, t+\Delta t \mid \mathbf{x}, t) \approx \mathcal{N}\left(\mathbf{x} + \mathbf{f}(\mathbf{x})\Delta t, \, \mathrm{diag}(g_1(\mathbf{x}), \ldots, g_d(\mathbf{x}))\Delta t\right). \tag{21}$$

More explicitly, the transition density is

$$p(\mathbf{x}', t+\Delta t \mid \mathbf{x}, t) = \frac{1}{(2\pi\Delta t)^{d/2}\prod_{i=1}^d \sqrt{g_i(\mathbf{x})}} \exp\left(-\frac{1}{2\Delta t}\sum_{i=1}^d \frac{(x'_i - x_i - \mathbf{f}_i(\mathbf{x})\Delta t)^2}{g_i(\mathbf{x})}\right), \tag{22}$$

whose logarithm reads

$$\log p(\mathbf{x}', t+\Delta t \mid \mathbf{x}, t) = -\frac{d}{2}\log(2\pi\Delta t) - \frac{1}{2}\sum_{i=1}^d \log g_i(\mathbf{x}) - \frac{1}{2\Delta t}\sum_{i=1}^d \frac{(x'_i - x_i - \mathbf{f}_i(\mathbf{x})\Delta t)^2}{g_i(\mathbf{x})}. \tag{23}$$

# C Synthetic Data Generation Model

In this section we provide more details about the generation of synthetic training data for `FIM-SDE`. First, we describe a scheme to sample SDEs of up to dimension $d_{\max}$, using polynomial coefficients of restricted degrees. Then, we specify how we sample from these SDEs to generate paths, which are potentially corrupted by spatial or temporal noise. Finally, we summarize the explicit hyperparameter choices made for the train set of `FIM-SDE`.

## C.1 Sampling Scheme for Polynomials

We begin with our sampling scheme for $n$-variate polynomials with degree bounded by $m_{\max}$. Below, this scheme is deployed to generate drift and diffusion functions that define the stochastic systems of our training data distribution.

A polynomial is uniquely defined by a finite set of monomials with non-zero coefficients. Hence, our sampling scheme derives such set of monomials (using a hierarchical sampling approach) and then samples (*a.s.*) non-zero coefficients for them. More precisely, our algorithm for sampling a $n$-variate polynomial $f$ is :

1. Sample the size of the set of degrees $N_{\text{deg}} \sim \mathcal{U}[1, \ldots, m_{\max}]$ with terms of non-zero coefficients in the polynomial.

2. Sample[7] $\{m_1, \ldots, m_{N_{\text{deg}}}\} \sim \mathcal{U}[\mathcal{P}_{N_{\text{deg}}}[\{0, \ldots, m_{\max}\}]]$, the set of monomial degrees with terms of non-zero coefficients in the polynomial.

3. For each $i \in \{1, \ldots, N_{\text{deg}}\}$, sample $N_{\text{mon}}^i \sim \mathcal{U}[1, \ldots, \binom{m_i + n - 1}{n - 1}]$, the number of monomials of degree $m_i$ with non-zero coefficients in the polynomial.

4. For each $i \in \{1, \ldots, N_{\text{deg}}\}$, sample $\{\boldsymbol{\alpha}_1^i, \ldots, \boldsymbol{\alpha}_{N_{\text{mon}}^i}^i\} \sim \mathcal{U}[\mathcal{P}_{N_{\text{mon}}^i}[\{\boldsymbol{\alpha} \in \mathbb{N}^n \mid |\boldsymbol{\alpha}| = m_i\}]]$, the $n$-variate exponents of monomials with non-zero coefficients in the polynomial.

5. For each $i \in \{1, \ldots, N_{\text{deg}}\}$ and $j \in \{1, \ldots, N_{\text{mon}}^i\}$ sample a coefficient $c_{\boldsymbol{\alpha}_j^i} \sim \mathcal{N}(0, 1)$ of the $n$-variate monomial $\mathbf{x}^{\boldsymbol{\alpha}_j^i}$.

6. The $n$-variate polynomial $f$ is then defined as $f(\mathbf{x}) = \sum_{i=1}^{N_{\text{deg}}} \sum_{j=1}^{N_{\text{mon}}^i} c_{\boldsymbol{\alpha}_j^i} \mathbf{x}^{\boldsymbol{\alpha}_j^i}$.

The uniform distributions cover a broad range of polynomials with degree bounded by $m_{\max}$, while the hierarchical sampling scheme ensures some sparsity (regarding monomials with non-zero coefficients) and correlation between monomials of the same degree.

As an example for $n = 3$ and $m_{\max} = 3$ consider the samples $N_{\text{deg}} = 2$, $\{m_1, m_2\} = \{0, 3\}$, with $N_{\text{mon}}^1 = 1$, $\{\boldsymbol{\alpha}_1^1\} = \{(0, 0, 0)\}$ and $N_{\text{mon}}^2 = 2$, $\{\boldsymbol{\alpha}_1^2, \boldsymbol{\alpha}_2^2\} = \{(3, 0, 0), (1, 1, 1)\}$. Then, for given $c_{\boldsymbol{\alpha}_1^1}, c_{\boldsymbol{\alpha}_1^2}, c_{\boldsymbol{\alpha}_2^2} \sim \mathcal{N}(0, 1)$, the sampled polynomial is $f(x_1, x_2, x_3) = c_{\boldsymbol{\alpha}_1^1} + c_{\boldsymbol{\alpha}_1^2} x_1^3 + c_{\boldsymbol{\alpha}_2^2} x_1 x_2 x_3$.

## C.2 Distribution over Drift Functions

To sample the drift function of a $d$-dimensional system, where $d \leq d_{\max}$, we apply the sampling scheme from Appendix C.1 for each component separately, using $n = d$ and $m_{\max} = m_{\max}^{\text{drift}}$. In other words, our distribution $p_{\text{drift}}(\mathbf{f}(\mathbf{x}) \mid d)$ factorizes over the $d$ components of $\mathbf{f}$ as $p_{\text{drift}}(f_1(\mathbf{x})) \ldots p_{\text{drift}}(f_d(\mathbf{x}))$. Each $f_i$ is sampled from the distribution over $d$-variate polynomials with degree bounded by $m_{\max}^{\text{drift}}$, specified by the algorithm in Appendix C.1.

## C.3 Distribution over Diffusion Functions

To define the distribution $p_{\text{diff}}(\mathbf{G}(\mathbf{x}) \mid d)$ over the diffusion functions of a $d$-dimensional system, where $d \leq d_{\max}$, recall from (4) that we only consider diagonal $\mathbf{G}(\mathbf{x})$. That is, for all $i \leq d$ there exists a non-negative function $g_i$ such that $\mathbf{G}(\mathbf{x})_{ii} = \sqrt{g_i(\mathbf{x})}$, and $\mathbf{G}(\mathbf{x})_{ij} = 0$ for all $j \neq i$. Hence, our distribution $p_{\text{diff}}(\mathbf{G}(\mathbf{x}) \mid d)$ also factorizes over the $d$ diagonal components as $p_{\text{diff}}(g_1(\mathbf{x}) \ldots p_{\text{diff}}(g_d(\mathbf{x}))$.

---

[7]For a set $S$ and $k \in \mathbb{N}$ we denote by $\mathcal{P}_k[S]$ the set of subsets of $S$ with $k$ elements.

We define $p_{\text{diff}}(g_i(\mathbf{x}))$ via an extension of the sampling scheme in Appendix C.1. First we sample $\tilde{g}_i$, a $d$-variate polynomial with degree bounded by $m_{\text{deg}}^{\text{diff}}$, using the sampling scheme from Appendix C.1. Because $\tilde{g}_i$ can attain negative values, we define the component function by $g_i(\mathbf{x}) = \max(0, \tilde{g}_i(\mathbf{x}))$. This then defines the component-wise distribution $p_{\text{diff}}(g_i(\mathbf{x}))$ and therefore, by our assumed factorization, $p_{\text{diff}}(\mathbf{G}(\mathbf{x}) \mid d)$.

Note that some diffusion component functions $g_i(\mathbf{x})$ sampled by this procedure contain regions with low, or even zero diffusion (*e.g.* when $\tilde{g}_i$ is a constant of negative value). In practice, we found that our trained model can therefore also approximate (almost) deterministic systems.

### C.4 SDE Simulation

To simulate paths from an SDE, with drift and diffusion functions sampled as described in Appendix C.2 and C.3, we first sample a set of $K$ initial states from $\mathcal{N}(0, 1)$. We solve the equation (i.e. sample paths) beginning at these $K$ *initial states* using the Euler–Maruyama method with a *fine discretization step size* $\Delta t$ until a *time horizon* $T$.

To facilitate smooth training, we employ two criteria to reject the equation, should (at least) one of them be satisfied. If an equation is rejected, we sample a new equation and repeat the simulation process until a predefined number of valid equations has been reached.

The first rejection criterion targets stability. It rejects an equation if at least one of the $K$ solutions contain NaN or Inf values. The second rejection criterion targets diverging systems. It rejects an equation if at least one component of at least one of the $K$ solutions exceeds a *threshold value* $\pm B$.

### C.5 SDE Corruption

In this subsection we describe how we implement the distribution $p_{\text{grid}}(\tau_1, \ldots, \tau_l)$ (over the *sequence recording times*) and $p_{\text{noise}}(\mathbf{y}_i | \mathbf{x}(t), \mathbf{f}, \mathbf{G})$ (over the optional *corruption* of the process values).

**Regular Observation Grids.** By default, we consider regular, and potentially coarse, observation grids. To realize such observations of our sampled equations, we subsample the fine grid simulations from Appendix C.4 regularly. We vary the subsampling factor, and therefore the *Regular inter-observation gap* $\Delta\tau$, throughout the dataset, to cover a wide range of application. Each subsampling yields $L$ remaining *observations per path*, still with a time horizon of $T$. See Appendix C.6 for more details about the specific choices we made for our dataset.

**Irregular Observation Grids**. To accommodate applications with *irregular inter-observation gaps*, we subsample these regular grids with an additional sampling scheme using the Bernoulli distribution. Given the regular, coarse observations of a process, we sample a single Bernoulli *survival probability* $\eta$ per equation. The irregular observation grid for a given equation are then realized by sampling the Bernoulli distribution with survival probability $\eta$ at each observation of all paths.

**Additive Gaussian Noise.** To make the model more robust, e.g. for real world applications, we corrupt the (clean) observations $\{\mathbf{x}_i\}_{i=1}^{K \times L}$ of a process $(\mathbf{f}, \mathbf{G})$ by additive Gaussian noise. The standard deviation is determined *relative to the observed values*. Concretely, let us define the component-wise *range* of the process:

$$r(\mathbf{x}(t), \mathbf{f}, \mathbf{G}) = \frac{1}{2} \left[ \left( \max_{i=1,\ldots,K \times L} \mathbf{x}_i \right) - \left( \min_{i=1,\ldots,K \times L} \mathbf{x}_i \right) \right] \tag{24}$$

To realize the additive Gaussian noise, we first sample a *noise scale* $\sigma(\mathbf{x}(t), \mathbf{f}, \mathbf{G})$ for the given (clean) observations of a process. Then, each observation is corrupted by samples from $\mathcal{N}(0, \sigma(\mathbf{x}(t), \mathbf{f}, \mathbf{G})r(\mathbf{x}(t), \mathbf{f}, \mathbf{G}))$. In other words:

$$p_{\text{noise}}(\mathbf{y}_i | \mathbf{x}(t), \mathbf{f}, \mathbf{G}) = \mathcal{N}(\mathbf{x}_i, \sigma(\mathbf{x}(t), \mathbf{f}, \mathbf{G})r(\mathbf{x}(t), \mathbf{f}, \mathbf{G})) \tag{25}$$

Note that a Course and Nair (2023) employ a similar relative additive noise scheme in some of their experiments on synthetic data. However, they define the range of a process based on the associated *ODE process*, i.e. with zero diffusion.

Table 4: The inter-observation times $\Delta\tau$, the number of paths $K$, the number of observations per path $L$ and the time horizon $T$ present in our dataset, and the percentage of data they occupy. Note that we vary the sampling step size $\Delta t$ as well, to speed up generation of paths with longer time horizon.

| % of Data | $\Delta t$ | $\Delta\tau$ | $K$ | $L$ | $K \times L$ | $T$ |
|---|---|---|---|---|---|---|
| 1/3 | 0.004 | 0.1 | 100 | 128 | 12800 | 12.8 |
| 1/3 | 0.002 | 0.01 | 25 | 512 | 12800 | 5.12 |
| 1/3 | 0.001 | 0.001 | 12 | 1024 | 12288 | 1.024 |

## C.6 Data Generation Hyperparameters

Let us now summarize the data generation hyperparameters of the train dataset of FIM-SDE.

We sample SDEs of up to dimension $d_{\max} = 3$. The drift function components are polynomials of up to degree $m_{\max}^{\text{drift}} = 3$. The diffusion function components are based on polynomials of up to degree $m_{\max}^{\text{diff}} = 2$.

We vary regular inter-observation times $\Delta\tau$, the number of paths $K$, the number of observations per path $L$ and the time horizon $T$ throughout the dataset. The information is summarized in Table 4. Note that we cover three orders of magnitude for $\Delta\tau$ and adapt $\Delta t$ for slightly faster generation. Moreover, we set $K$, $L$ and $T$ such that the number of observations per equation $K \times L$ is (roughly) fixed.

To corrupt the observation grids, we sample the Bernoulli survival probability as $\eta \sim \mathcal{U}[0.9, 1]$ per equation. The additive Gaussian noise scale is sampled as $\sigma(\mathbf{x}(t), \mathbf{f}, \mathbf{G}) \sim \mathcal{U}[0, 0.1]$ per equation. Each corruption scheme is applied to a third of the total dataset. Note that these corruptions overlap for a ninth of the total dataset, i.e. observations of a process can be noisy and on an irregular grid.

The rejection threshold value is set to $B = 100$, which leads to a rejection rate of $50\%$ for 1D, $78\%$ for 2D and $92\%$ for 3D equations. Higher dimensional equations therefore require more equation samplings, to populate the training distribution with valid equations, illustrating a *curse-of-dimensionality* effect. Moreover, higher dimensional equations are also (intuitively) more challenging to estimate.

We hence generate 300K 3D equations, 200K 2D equations and $100k$ 1D equations for our synthetic *train* dataset, for a total of 600K equations. An additional *validation* dataset of size 60K is generated using the same ratios.

# D FIM-SDE Model Details

This section contains the implementation details of FIM-SDE. We first describe a model input pre-processing and output post-processing approach that allows the model to be used a wide range of applications. We also provide more insights into the model architecture, training procedure and hyperparameters.

## D.1 Instance Normalization and Vector Field Renormalization

Observation sequences on different dynamical process are naturally characterized by different spatial and temporal length scales. To handle such sequences in a consistent manner, we introduce *component-wise instance normalization transformations*. We pre-process the inputs to our model before application, and post-process the outputs according to relevant change of variables formulas.

Continuing the notation from Section 4.2, let the data $\tilde{\mathcal{D}} = \{\mathbf{y}_i, \Delta\mathbf{y}_i, \Delta\mathbf{y}_i^2, \Delta\tau_i\}_{i=1}^{K(L-1)}$ and a location $\mathbf{x}$ be the inputs of FIM-SDE. We consider linear normalization transformations, so the normalizations of $\Delta\mathbf{y}$, $\Delta\mathbf{y}^2$ and $\Delta\tau$ are implied by the normalization transformations of $\mathbf{y}$ and $\tau$.

**Spatial Instance Normalization.** Before applying FIM-SDE, we *standardize* the observations $\{\mathbf{y}_i\}_{i=1}^{K(L-1)}$ *component-wise*. Let $y_{ij}$ denote the $j$-th component of $\mathbf{y}_i$, then the mean and standard

deviation of the $j$-th spatial dimension are

$$\bar{y}_j = \frac{1}{K(L-1)} \sum_{i=1}^{K(L-1)} y_{ij} \quad \text{and} \quad s_j = \sqrt{\frac{1}{K(L-1)} \sum_{i=1}^{K(L-1)} (y_{ij} - \bar{y}_j)^2} \quad . \tag{26}$$

These define $\bar{\mathbf{y}} = (\bar{y}_1, \ldots, \bar{y}_d) \in \mathbb{R}^d$ and $\mathbf{S} = \mathrm{diag}(s_1, \ldots, s_d) \in \mathbb{R}^{d \times d}$, which specify the *spatial transformation*

$$\mathcal{S}(\mathbf{y}) = \mathbf{S}^{-1}(\mathbf{y} - \bar{\mathbf{y}}) \tag{27}$$

and its inverse

$$\mathcal{S}^{-1}(\mathbf{y}) = \mathbf{S}\mathbf{y} + \bar{\mathbf{y}} \quad . \tag{28}$$

It is important to apply *the same* standardization transformation to the location $\mathbf{x}$, because they are elements of the now transformed domain.

**Temporal Instance Normalization.** The absolute time $\tau_i$ of an observation is not an input of our model. We therefore normalize $\Delta\tau_i$ directly. However, during application (as well as some part of our training data), $\Delta\tau_i$ might be constant, i.e. the input observations are on a regular grid. Moreover, $\Delta\tau_i$ might differ (vastly) between applications, even if all of them are on a regular grid. We therefore only *centralize* $\{\Delta\tau_i\}_{i=1}^{K(L-1)}$ around a target value, while keeping the inter-observation gaps positive for interpretability.

Let $\Delta\tau_{\mathrm{tar}} = 0.01$ be a target inter-observation gap after normalization and

$$\bar{\ln}_{\Delta\tau} = \frac{1}{K(L-1)} \sum_{i=1}^{K(L-1)} \ln \Delta\tau_i \tag{29}$$

the mean of $\{\ln \Delta\tau_i\}_{i=1}^{K(L-1)}$. Then we normalize the inter-observation gaps $\Delta\tau_i$ with

$$\Delta\tau_{\mathrm{tar}} \exp(-\bar{\ln}_{\Delta\tau})\Delta\tau_i \quad . \tag{30}$$

In other words, we center $\{\ln \Delta\tau_i\}_{i=1}^{K(L-1)}$ at $\ln \Delta\tau_{\mathrm{tar}}$.

Note that the unique corresponding *temporal transformation* $\mathcal{T}$ of absolute time $\tau$ that retains $0$, i.e. $\mathcal{T}(0) = 0$, is

$$\mathcal{T}(\tau) = \Delta\tau_{\mathrm{tar}} \exp(-\bar{\ln}_{\Delta\tau})\tau \tag{31}$$

with inverse

$$\mathcal{T}^{-1}(\tau) = \frac{\tau}{\Delta\tau_{\mathrm{tar}} \exp(-\bar{\ln}_{\Delta\tau})} \quad . \tag{32}$$

**Inverse Process Transformation.** `FIM-SDE` processes instance normalized inputs and therefore estimates a process in the *normalized domain*. To yield the corresponding process in the original *unnormalized domain*, we apply change of variables formulas according to the inverse of the spatial and temporal transformations from (28) and (32).

Let $\hat{\mathbf{f}}_\theta(\mathcal{S}(\mathbf{x})|\tilde{\mathcal{D}})$ and $\hat{\mathbf{G}}_\theta(\mathcal{S}(\mathbf{x})|\tilde{\mathcal{D}})$ denote the estimated drift and diagonal diffusion functions at the transformed location $\mathcal{S}(\mathbf{x})$. Recall that by our assumption in (4), the estimated diffusion matrix is diagonal.

According to Ito's formula and Theorem 8.5.7 in Øksendal (2010), the corresponding vector fields in the unnormalized domain at location $\mathbf{x}$ are

$$\Delta\tau_{\mathrm{tar}} \exp(-\bar{\ln}_{\Delta\tau})\mathbf{S}\hat{\mathbf{f}}_\theta(\mathcal{S}(\mathbf{x})|\tilde{\mathcal{D}}) \quad \text{and} \quad \sqrt{\Delta\tau_{\mathrm{tar}} \exp(-\bar{\ln}_{\Delta\tau})}\mathbf{S} \odot \hat{\mathbf{G}}_\theta(\tilde{\mathbf{x}}|\tilde{\mathcal{D}}) \quad , \tag{33}$$

where $\odot$ denotes the element-wise product.

### D.2 Model Architecture

In this section, we provide some more details about the architecture of the single `FIM-SDE`, all presented experiments were conducted with.

We fix a hidden size $n = 256$ for all embeddings throughout the different parts of the model.

All *features* are embedded with linear layers $\phi_\cdot^\theta$ to dimension $\frac{n}{4}$, s.t. their concatenated embedding $\mathbf{d}_i^\theta$ is of the desired size $n$. The embedded features are further encoded by the Transformer encoder with linear attention $\Psi^\theta$. It consists of 2 layers and attention dimension $n$.

Each *location* is a single input, encoded by another linear layer $\Phi_x^\theta$ to $n$ dimensions.

The trunk-net equivalent *functional attention mechanism* first applies 8 attention blocks (including residual feed-forward layers). The final embeddings $\mathbf{h}_{:,M}^\theta(\mathbf{x}|\tilde{D})$ are then projected to output dimension 3 by feed-forward networks with 2 hidden layers of dimension $n$.

Following standard conventions, the hidden size of the residual feed-forward neural networks in all transformer layers is set to $4n = 1024$.

We use a dropout of $0.1$ and the GeLU activation function.

In total, FIM-SDE has roughly 20M parameters.

## D.3 Training Procedure

We train FIM-SDE with AdamW (Loshchilov and Hutter, 2017), using learning rate $1e^{-5}$ and weight decay $1e^{-4}$.

In each batch, we sample the total number of observations passed to the model from $\mathcal{U}[128, 12800]$, so the model is exposed to widely varying context sizes. For each equation in a batch, we randomly sample 32 locations to compute the loss $\mathcal{L}$ from (10).

Memory requirements during training are quite large, as we use up to 12800 observations (inputs) per instance in a batch. We utilize four A100 40GB GPUs to train with a batch size of 64 for 1.3M optimization steps over roughly 6 days.

## D.4 Finetuning

In this section, we provide the explicit expressions we use to finetune FIM-SDE to some target data. Our finetuning approach depends on the assumed sampling rate of the observations.

**Dense Observations.** For densely sampled data, transitions between observations are approximately Gaussian (see Eq. (18)). We compute the log-likelihood of the short-time transitions induced by the estimators $(\hat{\mathbf{f}}_\theta, \hat{\mathbf{G}}_\theta)$, with respect to a target sequence $\mathbf{y}_1, \ldots, \mathbf{y}_L \in \mathbb{R}^d$ with small inter-observation times $\Delta\tau_l$. Specifically, we write

$$\mathcal{L}_{FT}^\theta = \sum_{l=1}^{L-1} \Big[ \sum_{i=1}^{d} \frac{(y_{l,i+1} - y_{l,i} - \hat{f}_{i,\theta}(\mathbf{y}_l)\Delta\tau_l)^2}{2\hat{g}_{i,\theta}(\mathbf{y})\Delta\tau_l} + \frac{1}{2}\log\hat{g}_{i,\theta}(\mathbf{y}) \Big]. \tag{34}$$

**Sparse Observations.** For sparsely sampled data, transitions between observations are not well approximated by a Gaussian. We alternatively simulate the one-step transition with the estimated equation and compare its result to the observed transition. Concretely, for an observed transition from $\mathbf{y}_l$ to $\mathbf{y}_{l+1}$ in time $\Delta\tau_l$, let $\hat{\mathbf{y}}_{l+1,\theta}$ be the result of a simulation of $(\hat{\mathbf{f}}_\theta, \hat{\mathbf{G}}_\theta)$ from initial state $\mathbf{y}_l$ for $\Delta\tau_l$. The finetuning objective is the mean squared error[8] between the ground-truth and estimated transition:

$$\mathcal{L}_{FT}^\theta = \frac{1}{L-1} \sum_{l=1}^{L-1} \|\mathbf{y}_{l+1} - \hat{\mathbf{y}}_{l+1,\theta}\|^2. \tag{35}$$

In both cases, the objective can be computed in parallel for all observations in the train split of the target data. Alternatively, the objective can be computed for a randomly sampled set of observations, if the set is sampled anew each iteration.

---

[8] The MSE could be replaced by other distance metrics, such as a Gaussian negative log-likelihood, under the assumption of additive noise. The variance of said noise term could also be learned.

# E   Baseline Models

Throughout our experiments, we compare `FIM-SDE` to three baseline models: `SparseGP`, `BISDE` and `LatentSDE`. This section offers a brief introduction to each method, provides relevant references and summarizes our hyperparameter choices.

## E.1   SparseGP

We use the `SparseGP` approach of Batz et al. (2018) as a nonparametric baseline for stochastic differential equation (SDE) estimation. In this method, both the drift and diffusion functions are modeled with Gaussian processes (GPs), and GP regression is performed directly on the observed one-step increments $(X_{t+\Delta t} - X_t)/\Delta t$ and their squared values $(X_{t+\Delta t} - X_t)^2/\Delta t$. This enables simultaneous, data-driven estimation of the deterministic and stochastic components of the dynamics without specifying any parametric form. We treat the observation interval $\Delta\tau$ as equivalent to the discretization step $\Delta t$, effectively assuming dense data, and always perform GP regression for both drift and diffusion.

For larger datasets, where direct GP regression becomes computationally expensive, we employ the sparse approximation proposed in the original work. In this setting, a fixed number of inducing points—chosen in regions of high trajectory density—are used to approximate the full GP posterior, significantly reducing computational cost while retaining accuracy. In our implementation, we use polynomial kernels for the drift functions and RBF kernels for the diffusion functions, with roughly ten inducing points per experiment. Kernel hyperparameters and diffusion noise levels are selected through evidence optimization.

## E.2   BISDE

`BISDE` (Wang et al., 2022) is a symbolic regression method for the low-dimensional SDE inference problem. Using sparse Bayesian learning, it avoids binning the data, a common technique in symbolic regression approaches. Binning is a costly operation, which suffers from a curse of dimensionality and introduces approximation error.

The effectiveness of `BISDE`, compared to a standard (SINDy-like) symbolic regression approach, is demonstrated in by Wang et al. (2022) in a range of experiments.

We use the open-source implementation[9] of `BISDE`, released by the authors. As a symbolic regression based method, it requires a family of basis function. Our choices, per dimensionality of the equation, are:

**1D:** $1, \quad x, \quad x^2, \quad x^3, \quad \sin(x), \quad \exp(x)$

**2D:** $1, \quad x, \quad y, \quad x^2, \quad y^2, \quad xy, \quad x^3, \quad x^2y, \quad xy^2, \quad y^3, \quad \sin(x), \quad \sin(y),$
$\quad \exp(x), \quad \exp(y), \quad \exp(xy), \quad \sin(xy)$

## E.3   LatentSDE

`LatentSDE` (Li et al., 2020) is a neural model that learns a (prior) SDE in latent space from data via variational inference. The model is trained end-to-end on a train split of the target datasets by maximizing an ELBO objective using gradient descent.

We use the open-source implementation of `LatentSDE` released by the authors in the `torchsde`[10] package.

In all experiments, we use the `LatentSDE` setup and training hyperparameters from the Lorenz experiment described in Li et al. (2020, Section 9.10.2). We report them here only for completeness.

The GRU encoder has 1 layer with hidden size 100. The latent dimension is of size 4. Drift and diffusion functions are implemented by 1 layer MLPs, also with hidden size 100. The prior drift is time-inhomogenous. The diffusion consists of separate MLPs for each latent dimension. Latent equations are solved with a fixed discretization step size of 0.01. For the Gaussian emission model,

---

[9] https://github.com/HAIRLAB/BISDE
[10] https://github.com/google-research/torchsde

Table 5: Sizes of the evaluation datasets. We report the sizes of the train splits, which is used as context for `FIM-SDE`, and the reference splits, used for MMD computations. Here, $\Delta\tau$ is the regular inter-observation time gap, $P$ is the number of paths, $T$ is the length of each path and the last column contains the total number of time points.

(a) Train data sizes.

| Dataset | Dim. | $\Delta\tau$ | P | T | Total |
|---|---|---|---|---|---|
| Can. Sys. | 1, 2 | 0.002, 0.02 | 1 | 5000 | 5000 |
| Lorenz | 3 | 0.025 | 1024 | 41 | 41984 |
| Wind | 1 | 0.167 | 4 | 5240 | 20960 |
| Oil | 1 | 1 | 4 | 1584 | 6336 |
| Facebook | 1 | $1.8e^{-5}$ | 4 | 4992 | 19968 |
| Tesla | 1 | $1.8e^{-5}$ | 4 | 4992 | 19968 |

(b) MMD reference data sizes.

| Dataset | Dim. | $\Delta\tau$ | P | T | Total |
|---|---|---|---|---|---|
| Can. Sys. | 1, 2 | 0.002 | 100 | 500 | 50000 |
| Lorenz | 3 | 0.025 | 128 | 41 | 5248 |
| Wind | 1 | 0.167 | 10 | 524 | 5240 |
| Oil | 1 | 1 | 10 | 158 | 1580 |
| Facebook | 1 | $1.8e^{-5}$ | 10 | 499 | 4990 |
| Tesla | 1 | $1.8e^{-5}$ | 10 | 499 | 4990 |

the mean is implemented as a linear projection from latent space. The standard deviation is assumed to be diagonal with a fixed standard deviation of 0.01. All MLPs employ the `softplus` activation function.

We train for 5000 iterations, where the KL term of the ELBO objective is annealed linearly over the first 50 iteration. Each iteration is performed on the whole train set, i.e. gradient descent without minibatches. The learning rate of the Adam optimizer (Kingma, 2014) is set to 0.01 with a decay of 0.999 per iteration. Gradients are computed through the solver, without the adjoint method.

Observation values are standardized per dimension and the observation times are normalized to $[0, 1]$ before training. For evaluation, model outputs are rescaled accordingly.

# F   Metrics

## F.1   Maximum Mean Discrepancy

The Maximum Mean Discrepancy (MMD) is a non-parametric statistical test used to compare two probability distributions (Gretton et al., 2012). MMD is a kernel based method that measures the difference between two distributions, $P$ and $Q$, purely based on samples drawn from these distributions. The samples are mapped to high-dimensional feature space, where linear methods are effectively applied.

Signature kernels for time series are implemented in the KSIG library (Tóth et al., 2025). Using the RBF-kernel and 5 signature levels, we obtain a signature kernel $k(\cdot, \cdot)$. The MMD between two sets of samples $\{\mathbf{x}_i\}_{i=1}^n \sim P$ and $\{\mathbf{y}_j\}_{j=1}^n \sim Q$ can then by computed as

$$\text{MMD}^2(P, Q) \approx \frac{1}{n(n-1)} \left( \sum_{i \neq j} k(\mathbf{x}_i, \mathbf{x}_j) + \sum_{i \neq j} k(\mathbf{y}_i, \mathbf{y}_j) \right) - \frac{2}{n^2} \sum_{i,j} k(\mathbf{x}_i, \mathbf{y}_j). \quad (36)$$

In our experiments, detailed in Appendix G, we generate or hold-out a set of reference paths specifically for the MMD computation. With each model, we simulate a set of paths of the same size and with the same time horizon. For consistency, we use the first observation of each path in the reference set as the initial states for the simulation. The time horizon (and number of observations) of the simulated paths is also determined from the set of reference paths.

To simulate these paths, `BISDE` and `SparseGP` learn an equation from a separate train set, that does *not* include the set of reference paths for MMD computation. `FIM-SDE` uses the same train set as context to infer an equation. Finetuning of `FIM-SDE` is always performed on the same train set. Finally, `LatentSDE` is trained on the train set, but the posterior initial states have to be inferred from the reference paths. We use the learned *prior equation* from `LatentSDE` to simulate paths, starting at these posterior initial states.

### F.2 Mean-squared Error on Vector Fields

In some experiments, we have access to the ground-truth vector fields used to generate the data paths. This is leveraged to compute the mean-squared error between the estimated and the ground-truth vector fields on a pre-defined grid. We define a regular grid in a hypercube that surrounds the observations. The mean-squared error is reported separately for drift and diffusion.

## G Experiments

This section covers our experiments in great detail, including a description of our employed metrics and a dedicated subsection for each dataset. We provide a thorough description of each dataset, our experimental setup and offer additional results. Table 5 contains information about the number of observations, number of paths and inter-observation time gaps for each dataset.

### G.1 Canonical SDE Systems

#### G.1.1 Description

We collect a set of well known SDEs with accompanying initial conditions, which were previously used by Batz et al. (2018), Course and Nair (2023) and Wang et al. (2022) to benchmark inference performance of their models. We simulate paths from each of these equations to generate datasets with fine and coarse grid observations and with optional Gaussian noise.

Here, we summarize the selected processes $d\mathbf{x}$ (where $\mathbf{W}$ is a Wiener processes), the initial conditions $\mathbf{x}(0)$ and the source of the equation.

**Double Well:**
$$dx = 4(x - x^3)\, dt + \sqrt{\max(4 - 1.25x^2, 0)}\, dW(t) \tag{37}$$
with $x(0) = 0$, extracted from Batz et al. (2018).

**2D Synthetic System:**
$$dx_1 = (x_1 - x_2 - x_1 x_2^2 - x_1^3)\, dt + \sqrt{1 + x_2^2}\, dW_1(t) \tag{38}$$
$$dx_2 = (x_1 + x_2 - x_1^2 x_2 - x_2^3)\, dt + \sqrt{1 + x_1^2}\, dW_2(t) \tag{39}$$
with $\mathbf{x}(0) = (1.5, 1.5)$, extracted from Wang et al. (2022).

**Damped Linear Oscillator:**
$$dx_1 = -(0.1\, x_1 - 2.0\, x_2)\, dt + dW_1(t) \tag{40}$$
$$dx_2 = -(2.0\, x_1 + 0.1\, x_2)\, dt + dW_2(t) \tag{41}$$
with $\mathbf{x}(0) = (2.5, -5)$, extracted from Course and Nair (2023).

**Damped Cubic Oscillator:**
$$dx_1 = -(0.1\, x_1^3 - 2.0\, x_2^3)\, dt + dW_1(t) \tag{42}$$
$$dx_2 = -(2.0\, x_1^3 + 0.1\, x_2^3)\, dt + dW_2(t) \tag{43}$$
with $\mathbf{x}(0) = (0, -1)$, extracted from Course and Nair (2023).

**Duffing Oscillator:**
$$dx_1 = x_2\, dt + dW_1(t) \tag{44}$$
$$dx_2 = -(x_1^3 - x_1 + 0.35\, x_2)\, dt + dW_2(t) \tag{45}$$
with $\mathbf{x}(0) = (3, 2)$, extracted from Course and Nair (2023).

**Selkov Glycolysis:**
$$dx_1 = -(x_1 - 0.08\, x_2 - x_1^2 x_2)\, dt + dW_1(t) \tag{46}$$
$$dx_2 = (0.6 - 0.08\, x_2 - x_1^2 x_2)\, dt + dW_2(t) \tag{47}$$

with $\mathbf{x}(0) = (0.7, 1.25)$, extracted from Course and Nair (2023).

**Hopf Bifurcation:**

$$dx_1 = (0.5\,x_1 + x_2 - x_1(x_1^2 + x_2^2))\,dt + dW_1(t) \tag{48}$$

$$dx_2 = (-x_1 + 0.5\,x_2 - x_2(x_1^2 + x_2^2))\,dt + dW_2(t) \tag{49}$$

with $\mathbf{x}(0) = (2, 2)$, extracted from Course and Nair (2023).

We simulate a single path from each systems with the Euler-Maruyama method and discretization step size $\Delta t = 0.002$. This path is subsampled by a factor of 1 or 10 to achieve inter-observation gaps of $\Delta\tau = 0.002$ or $\Delta\tau = 0.02$. The number of observations is kept fix at 5000, yielding time horizons of $T = 1$ or $T = 10$ respectively. Each resulting trajectory is corrupted by relative additive Gaussian noise (as in Appendix C.5) with standard deviation $\rho = 0.0$ or $\rho = 0.05$.

For each system $d\mathbf{x}$, each inter-observation gap $\Delta\tau$ and each standard deviation $\rho$, we repeat this sampling and evaluation five times. These five repetitions of each setup therefore yield a standard deviation of the models' performance.

We generate 100 additional paths with inter-observation gap $\Delta\tau = 0.002$ and 500 observations from each system $d\mathbf{x}$, without additive Gaussian noise. These paths are used as reference to compute the MMD, as described in Appendix F.1.

Finally, we record the ground-truth vector fields of each system at 1024 locations in an axis-aligned regular hypercube, roughly surrounding the observations. These hypercubes are defined by the span $[\text{lower}_i, \text{upper}_i]$ they occupy in dimension $i$. More precisely, we use $\{[-2, 2]\}$ for *Double Well*, $\{[-4, 4], [-4, 4]\}$ for *2D Synthetic System*, $\{[-2, 2], [-2, 2]\}$ for *Damped Linear Oscillator*, $\{[-2, 2], [-2, 2]\}$ for *Damped Cubic Oscillator*, $\{[-4, 4], [-4, 4]\}$ for *Duffing Oscillator*, $\{[-2, 4], [-2, 4]\}$ for *Selkov Glycolysis*, and $\{[-2, 2], [-2, 2]\}$ for *Hopf Bifurcation*.

### G.1.2 Experimental Setup

For each system $d$, each $\Delta\tau$ and $\rho$ and each of the five repetitions, the data generation yields a single path with 5000 observations. These are used as context by `FIM-SDE`, `BISDE` and `SparseGP` to estimate a drift and a diffusion function, specifying an SDE.

We sample paths from these estimated equations to compute the MMD. More precisely, we take the first observation of each of the 100 reference paths as initial states and simulate the equations on the *same observation grid as these reference paths*, i.e. with $\Delta\tau = 0.002$ and 500 observations. These two sets of 100 paths are used for the MMD computation (see Appendix F.1). The results are reported in Table 1 and include the standard deviation over the five repetition.

Moreover, we extract the values of the estimated vector fields in the 1024 locations of the regular hypercube. For each vector field, we compute the mean-squared error between the estimated and ground-truth values. Table 6 contains the results for the drift, Table 7 for the diffusion.

Process estimation or simulation can fail for the basline models `BISDE` and `SparseGP` in some setups or experiment repetitions. We remove those repetitions from the performance metric computation, and only compute the mean and standard deviation over the successful attempts. The number of failed repetitions are marked by Latin numerals in the respective tables.

### G.1.3 Additional Results

**Mean-squared Error on Vector Fields.** Tables 6 and 7 contain the mean-squared error between the ground-truth and estimated vector field values in a regular hypercube. In general, these results are complementary to the MMD results in Table 1. The baseline models perform decently on fine observation grids with no noise. `FIM-SDE` also handles coarser observation grids, potentially with additive Gaussian noise, very well.

**Figure of Vector Fields and Sample Paths.** Figure 5 contains ground-truth vector fields and sample paths of one experiment for each system. We display the results from the experimental setups without additive noise, $\rho = 0.0$, and the fine inter-observation grid, $\Delta\tau = 0.002$. For the vector fields, estimations from `SparseGP`, `BISDE` and `FIM-SDE` are displayed. We only show the sample paths from `FIM-SDE` for clarity. Vector field estimates and sample paths from `FIM-SDE` agree with the ground-truth.

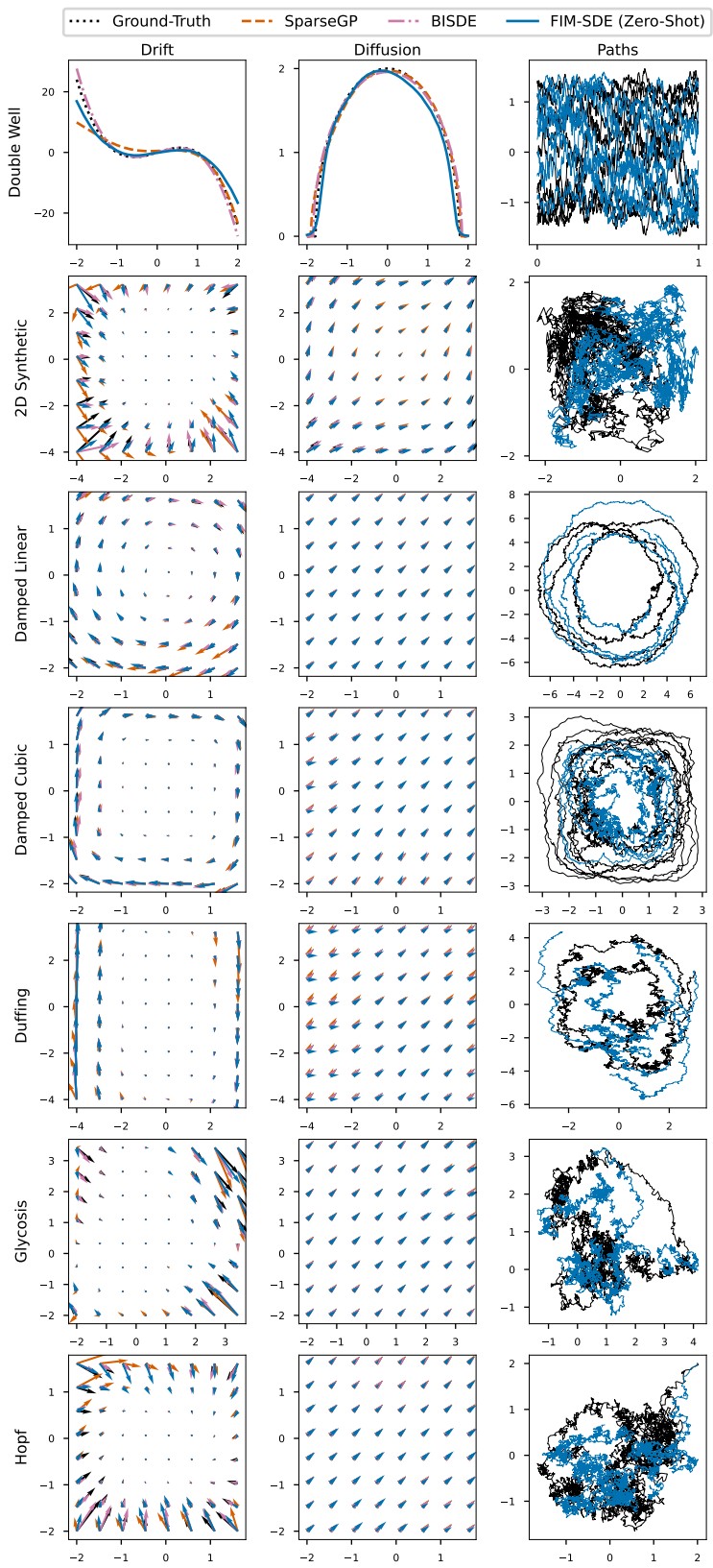

Figure 5: Drift and diffusion function estimation from `FIM-SDE` and baselines in all canonical SDE systems. The vector fields are estimated from observations with setup $\rho = 0.0$ and $\Delta\tau = 0.002$. The sample paths from `FIM-SDE` resemble the ground-truth paths closely.

Table 6: Mean-squared error on drift estimation in canonical SDE systems. The error is computed between values of the estimated and the ground-truth drift on the regular grid described in Appendix G.1.1. Drift functions estimated by `SparseGP` and `BISDE` might diverge, so we discard all values $> 10^4$ and indicate them by $-$.

| $\rho$ | $\Delta\tau$ | Model | Double Well | 2D-Synt (Wang) | Damped Linear | Damped Cubic | Duffing | Glycolysis | Hopf |
|---|---|---|---|---|---|---|---|---|---|
| 0.0 | 0.002 | SparseGP | 35.7 | 4750 | 3.92 | 1.39 | 1010 | 452 | 44.8 |
| | | BISDE | **4.94** | – | **0.203** | **0.461** | **1.56** | **49.0** | 10.5 |
| | | FIM-SDE | 8.34 | **1150** | 1.16 | 1.83 | 6.25 | 81.6 | **9.66** |
| | 0.02 | SparseGP | 21.2 | 621 | 3.47 | 478 | 3210 | 28.2 | 6.31 |
| | | BISDE | **1.92** | – | **0.0243** | 0.657 | **0.437** | **1.71** | **1.94** |
| | | FIM-SDE | 4.83 | **453** | 0.180 | **0.437** | 1.04 | 14.0 | 8.10 |
| 0.05 | 0.002 | SparseGP | 59.5 | – | 7.80 | 74.8 | 648 | – | 466 |
| | | BISDE | 819 | – | 1060 | 87.1 | 3380 | – | 4150 |
| | | FIM-SDE | **50.6** | **746** | **3.57** | **10.9** | **89.0** | **110** | **99.2** |
| | 0.02 | SparseGP | 33.0 | 546 | 3.69 | 8370 | 180 | 127 | 10.5 |
| | | BISDE | **0.409** | – | 1.23 | 2.67 | 21.0 | 45.1 | **6.50** |
| | | FIM-SDE | 2.31 | **489** | **0.297** | **0.591** | **5.87** | **30.1** | 6.77 |

Table 7: Mean-squared error on diffusion estimation in canonical SDE systems. The error is computed between values of the estimated and the ground-truth diffusion on the regular grid described in Appendix G.1.1.

| $\rho$ | $\Delta\tau$ | Model | Double Well | 2D-Synt (Wang) | Damped Linear | Damped Cubic | Duffing | Glycolysis | Hopf |
|---|---|---|---|---|---|---|---|---|---|
| 0.0 | 0.002 | SparseGP | 0.0227 | 1.70 | 0.0958 | 0.0205 | 0.152 | 0.0411 | 0.00373 |
| | | BISDE | **0.00620** | 0.548 | **0.000105** | **0.000115** | **0.000114** | **0.0264** | **0.00197** |
| | | FIM-SDE | 0.0209 | **0.349** | 0.0111 | 0.00619 | 0.0945 | 0.112 | 0.114 |
| | 0.02 | SparseGP | 0.205 | 1.37 | 0.0302 | 1.42 | 0.848 | 0.129 | 0.00749 |
| | | BISDE | 0.0460 | 1.26 | **0.0000661** | **0.000331** | **0.000325** | **0.00223** | **0.00484** |
| | | FIM-SDE | **0.0334** | **0.268** | 0.0177 | 0.127 | 0.128 | 0.123 | 0.106 |
| 0.05 | 0.002 | SparseGP | 3.21 | **2.18** | 115 | 10.5 | 89.5 | 7.04 | 4.26 |
| | | BISDE | 2.43 | 2440 | 75.8 | 10.2 | 72.8 | 7.38 | 3.96 |
| | | FIM-SDE | **1.26** | 2.19 | **2.87** | **0.825** | **2.33** | **0.224** | **0.252** |
| | 0.02 | SparseGP | 0.553 | 1.08 | 6.48 | 2.12 | 7.58 | 1.00 | 0.151 |
| | | BISDE | 0.0922 | **0.395** | 5.52 | 0.881 | 8.01 | 0.744 | **0.113** |
| | | FIM-SDE | **0.0371** | 0.733 | **2.07** | **0.334** | **2.50** | **0.171** | 0.129 |

## G.2 Lorenz

### G.2.1 Description

We generate synthetic data from a Lorenz attractor system with constant diagonal diffusion, following the setup from Li et al. (2020, Section 9.10.2). For completeness, we recall the setup in this section.

Li et al. (2020) consider the Lorenz system

$$
\begin{align}
dx_1 &= \sigma(x_2 - x_1)\, dt + \alpha\, dW_1(t) \tag{50}\\
dx_2 &= (x_1(\rho - x_3) - x_2)\, dt + \alpha\, dW_2(t) \tag{51}\\
dx_3 &= (x_1 x_2 - \beta x_3)\, dt + \alpha\, dW_3(t) \tag{52}
\end{align}
$$

with parameters $\sigma = 10$, $\rho = 28$, $\beta = 8/3$ and $\alpha = 0.15$. Initial states are sampled from the standard Gaussian distribution $\mathcal{N}(\mathbf{0}, \mathbf{I})$.

We simulate paths in the time interval $[0, 1]$ on a fine grid and record observations in a coarse regular grid with 41 points, corresponding to a inter-observation gap $\Delta\tau = 0.025$. These paths are

standardized in each dimension. Afterwards, they are corrupted with additive Gaussian noise using a standard deviation of $0.01$.

We generate a train set of size $1024$ and a validation set of size $128$, which is standardized using the statistics from the train set.

For the MMD computation, we generate three additional reference datasets with $128$ paths each, using three distinct initial distributions: $\mathcal{N}(\mathbf{0}, \mathbf{I})$, $\mathcal{N}(\mathbf{0}, 2\mathbf{I})$ and a fixed value of $(1, 1, 1)$. Simulation time interval and inter-observation gap remain at $[0, 1]$ and $\Delta\tau = 0.025$ respectively. Standardization is performed with the statistics from the train set. No additive Gaussian noise is applied.

### G.2.2 Experimental Setup

We compare `FIM-SDE` to `LatentSDE` with MMD, using the additional reference sets.

Implementing `LatentSDE` with the hyperparameters described in Appendix E.3, we train the model on the train split for $5000$ epochs. To sample paths for the MMD computation, the trained model encodes the reference set and estimates a posterior initial condition. Using the learned prior equation, we sample a set of $128$ paths on the same observation grid as the MMD reference paths.

`FIM-SDE` processes the train data as context and estimates an equation. Using the first values of each MMD reference path as initial states, we also sample a set of $128$ paths per reference dataset.

The MMD is computed between the set of reference paths and these sampled model paths. We report the results in Table 2.

### G.3 Real-World Systems

### G.3.1 Description

The set of four emirical datasets, representing complex real-world phenomena, were collected and studied by Wang et al. (2022). They were released alongside the open-source implementation of `BISDE`[11]. We provide a brief description of each of them and refer to Wang et al. (2022) for more details.

**Stock Prices:** The prices of *Facebook* and *Tesla* stock were recorded every minute from July to September 2020. As such, there are $24960$ one-dimensional observations per stock. Following Wang et al. (2022), the models process log-transformed observations. For evaluation, the estimated equations are transformed back to the original data space with Ito's formula.

**Wind Speed Fluctuation:** For this dataset, wind speeds at a meteorological station in Wellington, New Zealand, were recorded every ten minutes in the first half of 2020. The fluctuations, i.e. the difference between two consecutive wind speed recordings, are modeled. There are $26202$ one-dimensional observations in total.

**Oil Price Fluctuations:** Daily crude oil prices from 1986 to 2017, collected from data by the US Energy Information Administration. The daily price fluctuations are modeled, yielding $7921$ one-dimensional observations.

To obtain meaningful results and a standard deviation for the MMD metric, we perform five-fold cross-validation. Each dataset is split into five splits of equal size. Data from four splits is used as context for the models, while the last split provides reference paths. We split this reference split into ten paths of equal size, because MMD computation requires a set of reference paths.

### G.3.2 Experimental Setup

We compare `FIM-SDE` to `BISDE` and `LatentSDE` on each of the four datasets with MMD, using our five-fold cross-validation setup.

For each instance of the cross-validation, data from four splits is used as context for `FIM-SDE` and as training data for `BISDE` and `LatentSDE`. From the estimated (prior) equations, we sample ten paths on the same observation grid as the ten paths from the MMD reference split. `FIM-SDE` and `BISDE`

---

[11] https://github.com/HAIRLAB/BISDE

use the first observations of the reference paths as initial states. `LatentSDE` infers a posterior initial condition in latent space by encoding the ten reference paths.

The MMD is computed between the set of MMD reference paths and this set of ten sampled paths from the respective model. The five-fold cross-validation yields a mean and standard deviation of the MMD, aggregating the results per dataset and model.

Table 3 contains the results of this experiment. `FIM-SDE` performs better than `BISDE` in the two stock price datasets, but worse in the two fluctuation datasets. `LatentSDEs` performance is similar to `FIM-SDE`.

### G.3.3   Additional Results

`BISDE` **Equations.** We report the equations estimated by `BISDE` per dataset and cross-validation split in Table 8.

Table 8: Symbolic drift and diffusion function estimates from `BISDE` in real-world systems. We report the functions per cross-validation split for each real-world system.

| System | Split | Drift | Diffusion |
|---|---|---|---|
| Wind | 1 | $-7.2480\,x + 0.6628\,\sin(x)$ | $2.9061 - 1.5182\,x + 1.5683\,x^2 + 0.1565\,x^3 + 1.9734\,\sin(x) - 0.0746\,e^x$ |
| | 2 | $-7.1237\,x + 0.4295\,\sin(x)$ | $3.0896 - 1.3526\,x + 1.4672\,x^2 + 0.1223\,x^3 + 1.7693\,\sin(x) - 0.0651\,e^x$ |
| | 3 | $-6.9100\,x + 0.2030\,\sin(x)$ | $2.8466 - 2.4500\,x + 1.4562\,x^2 + 0.2513\,x^3 + 2.8331\,\sin(x) - 0.0877\,e^x$ |
| | 4 | $-6.8015\,x + 0.1054\,\sin(x)$ | $3.1170 + 1.3119\,x^2 + 0.3860\,\sin(x) - 0.0601\,e^x$ |
| | 5 | $-6.8845\,x + 0.1443\,\sin(x)$ | $2.8354 - 1.6006\,x + 1.5005\,x^2 + 0.1349\,x^3 + 2.0828\,\sin(x) - 0.0646\,e^x$ |
| Oil | 1 | $-1.0449\,x$ | $1.0427 + 0.3162\,x^2$ |
| | 2 | $-1.0447\,x$ | $1.1164 + 0.3060\,x^2$ |
| | 3 | $-1.0426\,x$ | $1.0448 + 0.3108\,x^2$ |
| | 4 | $-1.0405\,x$ | $0.5797 + 0.3132\,x^2 - 0.0909\,\sin(x)$ |
| | 5 | $-1.0426\,x$ | $0.7484 + 0.3346\,x^2$ |
| Facebook | 1 | $0$ | $0.1720$ |
| | 2 | $0$ | $0$ |
| | 3 | $0$ | $0$ |
| | 4 | $0$ | $-0.2075\,\sin(x)$ |
| | 5 | $-0.2555\,\sin(x)$ | $0$ |
| Tesla | 1 | $-0.8620\,\sin(x)$ | $0$ |
| | 2 | $-10.5399\,\sin(x)$ | $0$ |
| | 3 | $-9.1783\,\sin(x)$ | $1.3294$ |
| | 4 | $-7.2514\,\sin(x)$ | $0.6220$ |
| | 5 | $-6.5110\,\sin(x)$ | $0.4163\,\sin(x)$ |

**Convergence Speed.** We additionally compare the performance on the four real-world systems over the course of training of `LatentSDE` and finetuning of `FIM-SDE`.

The experimental setup remains the same: a five-fold cross-validation evaluated by the MMD to the reference split. The results are summarized in Table 9.

We observe that `LatentSDE` requires thousands of iterations to converge, while `FIM-SDE` reaches comparable or superior performance after only 100 iterations.

**Figure of Sample Paths.** Figure 6 contains the reference paths and estimations from `LatentSDE` and `FIM-SDE` for one cross-validation split of each system. In zero-shot mode, `FIM-SDE` performs roughly as well as the custom trained `LatentSDE`. For the oil and wind systems, finetuning `FIM-SDE` improves the sample path quality drastically. In the stock price data, finetuning does not affect the already convincing paths from the zero-shot model.

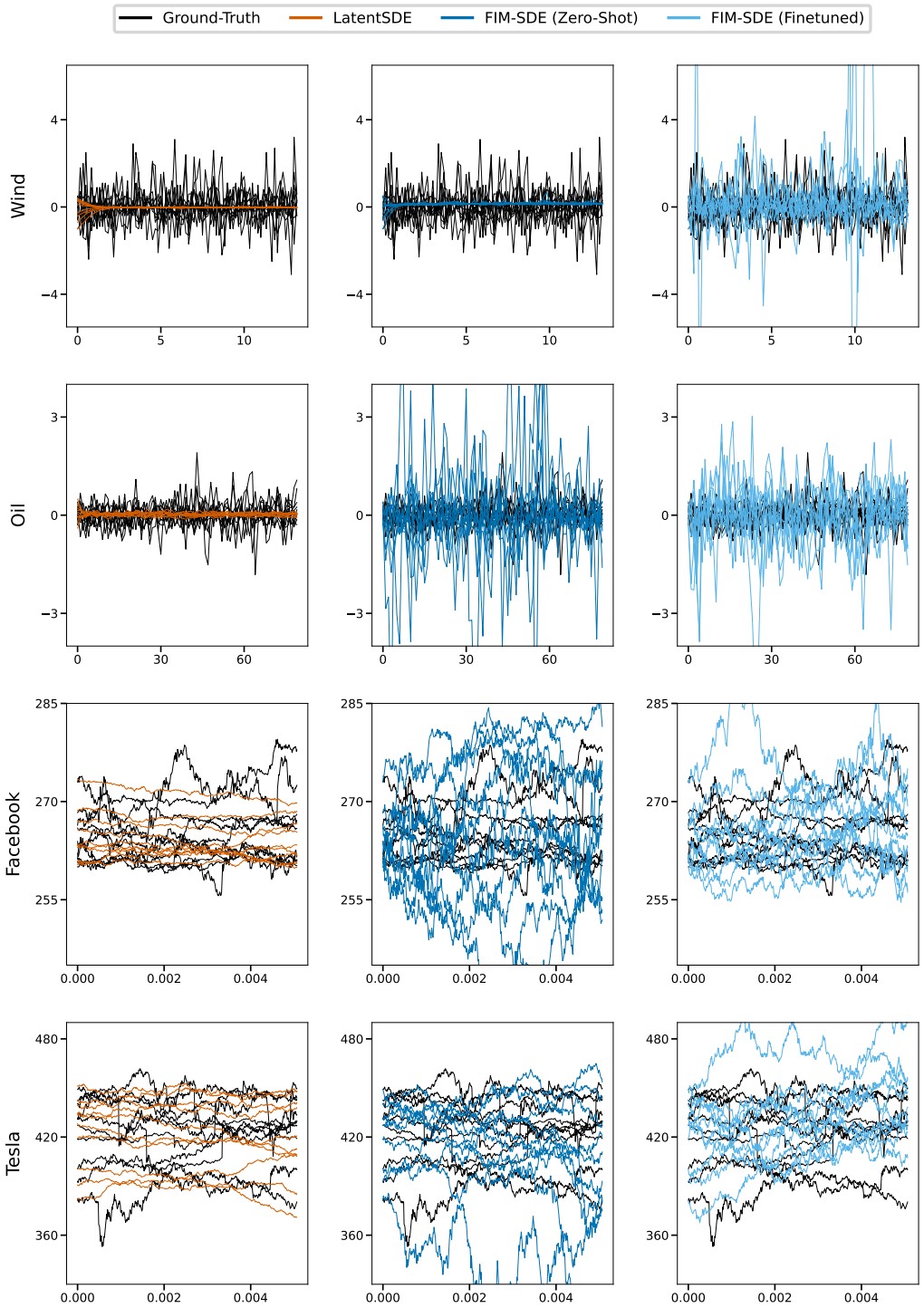

Figure 6: Paths from one cross-validation split in each real-world system. For each system we plot the reference paths of this split and the estimations of `LatentSDE` and `FIM-SDE`, in zero-shot mode and finetuned. The displayed trajectories for oil and wind are truncated for clarity.

Table 9: Convergence speed of training `LatentSDE` and finetuning `FIM-SDE` in four real-world systems. Depicted is mean and standard deviation of the MMD over five cross-validation splits per system at different training (resp. finetuning) iterations.

| Model | Iteration | Wind | Oil | Facebook | Tesla |
|---|---|---|---|---|---|
| `LatentSDE` | 500 | $1.019 \pm 0.078$ | $0.382 \pm 0.236$ | $0.091 \pm 0.043$ | $0.066 \pm 0.059$ |
| | 1000 | $0.874 \pm 0.127$ | $0.224 \pm 0.185$ | $0.071 \pm 0.037$ | $0.052 \pm 0.056$ |
| | 2000 | $0.318 \pm 0.164$ | $0.205 \pm 0.172$ | $0.034 \pm 0.035$ | $0.023 \pm 0.028$ |
| | 5000 | $0.350 \pm 0.212$ | $0.235 \pm 0.167$ | $0.014 \pm 0.018$ | $0.016 \pm 0.023$ |
| `FIM-SDE` (Zero-Shot) | $-$ | $0.330 \pm 0.11$ | $0.199 \pm 0.137$ | $0.012 \pm 0.026$ | $0.000 \pm 0.000$ |
| `FIM-SDE` (Finetuned) | 20 | $0.114 \pm 0.06$ | $0.071 \pm 0.094$ | $0.020 \pm 0.019$ | $0.021 \pm 0.014$ |
| | 40 | $0.075 \pm 0.05$ | $0.048 \pm 0.057$ | $0.019 \pm 0.026$ | $0.020 \pm 0.028$ |
| | 60 | $0.042 \pm 0.04$ | $0.049 \pm 0.034$ | $0.003 \pm 0.007$ | $0.013 \pm 0.018$ |
| | 80 | $0.043 \pm 0.07$ | $0.046 \pm 0.028$ | $0.002 \pm 0.005$ | $0.011 \pm 0.016$ |
| | 100 | $0.029 \pm 0.03$ | $0.068 \pm 0.042$ | $0.006 \pm 0.007$ | $0.004 \pm 0.008$ |

# H   Ablation Studies

In this section, we present ablation studies on the training and application of `FIM-SDE`. First, we investigate the scaling of `FIM-SDE`, by scaling the train data size and the number of model parameters. Then we consider the train data distribution, ablating the degree of polynomials it consists of. Lastly, we study the impact of the context size on the performance of `FIM-SDE`, exemplified by data from two canonical systems.

## H.1   Ablating Size of Train Data and Model Parameter Count

Table 10: Ablation study on scaling `FIM-SDE` in terms of parameter count and train data size. We report the MMD, scaled by a factor of 100, on canonical systems, which can be compared to the results from Table G.1.

| $\rho$ | $\Delta\tau$ | Param. Count | Data Size | Double Well | 2D-Synt (Wang) | Damped Linear | Damped Cubic | Duffing | Glycolysis | Hopf |
|---|---|---|---|---|---|---|---|---|---|---|
| 0.0 | 0.002 | 5M | 30k | 30(1) | 10(3) | 10(2) | 13(4) | 9(4) | 11(4) | 13(4) |
| | | 10M | 100k | **0.7(6)** | **8(3)** | **8(2)** | **7(3)** | **7(2)** | **10(4)** | **6.3(8)** |
| | 0.02 | 5M | 30k | **0.4(5)** | 2.5(5) | 0.5(5) | 3.8(6) | 0.6(2) | 1.5(6) | 3.2(3) |
| | | 10M | 100k | 0.5(7) | **0.2(4)** | **0.01(2)** | **0.6(4)** | **0.07(8)** | **0.3(2)** | **0.1(2)** |
| 0.05 | 0.002 | 5M | 30k | 30(1) | 10(4) | **10(2)** | **12(4)** | 16(8) | **12(7)** | 14(4) |
| | | 10M | 100k | **3.9(6)** | **7(3)** | 10(2) | **14(2)** | 14(3) | 13(4) | **9(2)** |
| | 0.02 | 5M | 30k | 0.8(6) | 1.4(6) | **1.1(7)** | 3.8(5) | 3(1) | 1.2(5) | 1.2(5) |
| | | 10M | 100k | **0.4(8)** | **0.4(8)** | 1.9(5) | **0.5(2)** | **2.9(10)** | **0.8(6)** | **0.0(0)** |

Recall from Appendix D that `FIM-SDE` has roughly 20M parameters and was trained on 600k equations. We train two smaller models on fewer equations to investigate the scaling behavior of `FIM-SDE`. One model has 10M parameters and is trained on 100k equations. The other, even smaller, model has 5M parameters and is trained on just 30k equations. All models are trained for 500k optimization steps.

We compare both models in terms of MMD on the set of canonical systems from Appendix G.1. The results are reported in Table 10 and can be compared to the results in Table 1.

Without relative additive noise, i.e. $\rho = 0.0$, the larger model trained on more data performs consistently better. Only in a few setups, results of the two models less than one standard deviation apart, otherwise the larger model is (significantly) better. Although less decisive, the same pattern emerges on data with relative additive noise, i.e. $\rho = 0.05$.

Note that both (smaller) models are worse than the (larger) `FIM-SDE` from Table 1. These results validate our choice of scaling the model and train data size for better overall performance.

Table 11: Ablation study on the train data distribution of `FIM-SDE`, in particular the maximal degree of drift polynomials in the train data. We report the MMD, scaled by a factor of 100, on canonical systems, which can be compared to the results from Table G.1.

| $\rho$ | $\Delta\tau$ | Drift Degree | Double Well | 2D-Synt (Wang) | Damped Linear | Damped Cubic | Duffing | Glycolysis | Hopf |
|---|---|---|---|---|---|---|---|---|---|
| 0.0 | 0.002 | 3 | 30(1) | 10(3) | 10(2) | 13(4) | 9(4) | 11(4) | 13(4) |
| | | 4 | **3(1)** | **7(2)** | **6(1)** | **6(2)** | **6(1)** | **10(3)** | **8(2)** |
| | 0.02 | 3 | **0.4(5)** | **2.5(5)** | 0.5(5) | 3.8(6) | 0.6(2) | 1.5(6) | 3.2(3) |
| | | 4 | **0.4(6)** | 2.7(7) | **0.1(1)** | **2.9(8)** | **0.33(10)** | **1.(5)** | **1.4(4)** |
| 0.05 | 0.002 | 3 | 30(1) | **10(4)** | 10(2) | **12(4)** | 16(8) | **12(7)** | 14(4) |
| | | 4 | **9(3)** | 11(4) | **9(2)** | 15(4) | **12(4)** | 16(5) | **10(2)** |
| | 0.02 | 3 | 0.8(6) | **1.4(6)** | **1.1(7)** | **3.8(5)** | 3(1) | **1.2(5)** | 1.2(5) |
| | | 4 | **0.2(2)** | 2.1(4) | 1.7(9) | 4.0(4) | **2.5(9)** | 2(5) | **1.1(5)** |

## H.2 Ablating the Train Data Distribution

In Appendix C we describe the synthetic train data distribution for `FIM-SDE`. Recall that this data is generated from SDEs with polynomial drift of up to degree 3. In this Section, we ablate the polynomial degree of the drift functions and study its impact.

We train two models with 5M parameters on 30k SDEs. For one model, the 30k SDEs are sampled with $m_{\max}^{\text{drift}} = 3$, as described in Appendix C.6. The other model is trained on 30k SDE using the same setup, apart from allowing degree 4 drift functions, i.e. $m_{\max}^{\text{drift}} = 4$.

We compare the two models on their performance in terms of MMD on the set of canonical systems from Appendix G.1 and report the results in Table 11.

The results often differ by less than one standard deviation. Without noise, i.e. $\rho = 0.0$, the model trained on up to degree 4 drift seem to perform slightly better than the other model, when only considering the mean performance. This advantage vanishes if noise is added, i.e. $\rho = 0.05$.

Considering the significantly higher sampling rejection rate when generating the data with degree 4 drift, while only providing minimal performance increases, we decided to train `FIM-SDE` on data with $m_{\max}^{\text{drift}} = 3$.

## H.3 Ablating the Context Size during Evaluation

During *training*, `FIM-SDE` has been exposed to context sizes from 128 to 12800 observations (see Appendix D). In this section, we ablate the context size of `FIM-SDE` during *evaluation* and study its impact on performance.

We conduct this ablation study on two canonical systems from Appendix G.1: the Double Well and the Damped Linear Oscillator. Sets of context paths of sizes ranging from 500 to 50000 are generated using the fine inter-observation gap $\Delta\tau = 0.002$ and no additive Gaussian noise $\rho = 0$. Using the experimental setup from Appendix G.1, we evaluate the performance of `FIM-SDE` on these different context sizes using the MMD and repeating each experiment five times.

In Table 12 we report the results of ablating the context size of `FIM-SDE` over two orders of magnitude. The model requires at least 2000 context points to infer the Damped Linear Oscillator correctly. The Double Well system, inherently more difficult because of its state-dependent diffusion function, requires 4000 context points. Increasing the context size, even beyond the train maximum of 12800, does not deteriorate the already good performance of `FIM-SDE`.

Table 12: Ablation study on the context size of `FIM-SDE`. We consider data from two canonical systems (see Appendix G.1), generate with $\rho = 0$ and $\Delta\tau = 0.002$. Ablating the number of observations, we report the mean and standard deviation of the MMD across five repetitions of each experiment.

| Context Size | Double Well | Damped Linear Oscillator |
|---:|:---:|:---:|
| 500 | $0.155 \pm 0.090$ | $0.128 \pm 0.068$ |
| 1000 | $0.115 \pm 0.094$ | $0.014 \pm 0.005$ |
| 2000 | $0.053 \pm 0.073$ | $0.006 \pm 0.005$ |
| 3000 | $0.042 \pm 0.086$ | $0.002 \pm 0.003$ |
| 4000 | $0.004 \pm 0.003$ | $0.000 \pm 0.000$ |
| 5000 | $0.004 \pm 0.003$ | $0.000 \pm 0.000$ |
| 50000 | $0.006 \pm 0.006$ | $0.000 \pm 0.000$ |

