# OpenReview forum: "In-Context Learning of Stochastic Differential Equations with Foundation Inference Models"
_NeurIPS.cc/2025/Conference — NeurIPS 2025 poster_

### Official Review · Reviewer_P9Vv · 2025-06-19

**Clarity:** 3
**Significance:** 3
**Originality:** 3
**Rating:** 4
**Confidence:** 3

**Summary:**

FIM-SDE introduces a pre-trained neural network capable of performing zero-shot estimations of drift and diffusion functions of SDEs from noisy, irregularly sampled time series data without requiring parameter fine-tuning for each new system. The approach leverages amortized inference and neural operator concepts by training a Transformer-based architecture on a large synthetic dataset of 600K low-dimensional SDEs with polynomial drift functions (up to degree 3) and diagonal diffusion matrices, where the model learns to map corrupted SDE observations to their underlying governing functions through a context matrix encoding and functional attention mechanism. FIM-SDE addresses key limitations of existing methods—heavy reliance on prior knowledge, sensitivity to noise, and computational challenges of neural SDE training—by providing a simulation-free solution that can generalize across diverse dynamical systems. Extensive evaluation of eight canonical SDE systems and four real-world datasets (including stock prices, oil prices, and wind speed fluctuations) demonstrates that FIM-SDE consistently outperforms traditional Gaussian process regression and symbolic methods, particularly under noisy conditions while offering the practical advantage of immediate deployment without system-specific optimization, positioning it as a foundation model for automated scientific discovery in stochastic dynamical systems.

**Questions:**

Can you clearly demonstrate some usecases for the paper? For instance, can you compare the training and regression computing tradeoffs between this approach and the on-the-go approaches?

**Ethical Concerns:**

["NO or VERY MINOR ethics concerns only"]

**Final Justification:**

My concerns are addressed by the author's detailed explanations and new experimental results, and will remain my positive feedback.

**Limitations:**

yes

**Quality:**

3

**Strengths And Weaknesses:**

Strengths:
1. This idea of a GPT-alike SDE fitting model is novel and interesting. While the performance against existing methods is not always stellar, this is a good first attempt.
2. The paper is well-written, very clear motivation for the paper.

Weakness:
1. Like mentioned above, the experimental results are not stellar always.
2. The computational cost seems really high. The idea of such a model is that at inference time this is faster and cheaper than other on the go model, but it is not so clear.

---

> ### Author Rebuttal · Authors · 2025-07-30
>
> We sincerely thank the reviewer for their time, for their comments and questions, and for their positive assessment of our submission. Below, we address each of the points raised.
>
> **@W1:** Let us begin by summarising the experiments we have prepared for the rebuttal into two tables:
>
> 1. *Real‑world data (Table 2a, extended)*. We added (i) FIM‑SDE fine‑tuned on each data set and (ii) LatentSDE [1] as an additional, neural network baseline.
>
> | Model           | Wind               | Oil                | Facebook           | Tesla                |
> |:----------------|:------------------:|:------------------:|:------------------:|:--------------------:|
> | BISDE           | $0.000 \\pm 0.000$ | $0.137 \\pm 0.097$ | $0.053 \\pm 0.060$ | $0.013 \\pm 0.015$   |
> | LatentSDE       | $0.350 \\pm 0.212$ | $0.235 \\pm 0.167$ | $0.014 \\pm 0.018$ | $0.016 \\pm 0.023$   |
> | FIM-SDE (zero-shot) | $0.330 \\pm 0.114$ | $0.199 \\pm 0.137$ | $0.012 \\pm 0.026$ | $0.000 \\pm 0.000$   |
> | FIM-SDE (fine-tuned)   | $0.029 \\pm 0.034$ | $0.068 \\pm 0.042$ | $0.006 \\pm 0.007$ | $0.004 \\pm 0.008$   |
>
>
> 2. *Stochastic Lorenz attractor (Table 2b, corrected)*. The original numbers were unfortunately computed using an incorrect normalization factor. The corrected scores below can be easily reproduced using the pretrained FIM-SDE model provided with our submission.
>
> | Model         | $\\mathcal{N}(0,1)$ | $(1, 1, 1)$ | $\\mathcal{N}(0, 2)$ |
> |:--------------|:--------------------|:------------|:---------------------|
> | LatentSDE     |  $0.051708$         |  $0.135003$ |  $0.054335$          |
> | FIM-SDE (zero-shot) |  $0.403933$         |  $1.151080$ |  $0.378515$          |
> | FIM-SDE (fine-tuned) |  $0.003838$         |  $0.037159$ |  $0.003335$          |
>
> Key take‑aways:
>
> - The **zero-shot** performance of FIM-SDE on real-world data is not only competitive with symbolic regression methods (BISDE)  but also with neural models (LatentSDE).
> - A short fine‑tune (${\\leq}\\ 100$ iterations) improves FIM‑SDE dramatically, surpassing LatentSDE, which needs ${\\approx}\\ 5\\ 000$ iterations to reach its reported scores (see **@W2** & **@Q1** below).
> -  The stochastic Lorenz attractor highlights a bias in our pre‑training distribution, as discussed in **@Q6** to reviewer WGed. Nevertheless, the same quick fine‑tune eliminates the gap and again outperforms LatentSDE.
>
> These results demonstrate that FIM-SDE provides a competitive zero-shot solution to a wide variety of dynamical systems. They also demonstrate that FIM-SDE can easily be fine-tuned to synthetic and real-world datasets and that the result outperforms neural SDE solutions.
>
> **@W2** & **@Q1**: Since the second weakness and the main question raised are related, we address them together in what follows.
>
> FIM‑SDE is intended as an *off‑the‑shelf* tool for scientists who have noisy trajectories but little ML expertise. Like all foundation models, **pre‑training is a one‑time, centralised expense**. End‑users download the checkpoint and either (i) use it zero‑shot, or (ii) fine‑tune briefly on their data. The figures below therefore *exclude* pre‑training cost and focus on the compute a user would spend per new data set.
>
> *Use-case Scenario*: You receive the four real‑world data sets of Section 4.2 and have the following methods available
>
> - BISDE [3]: symbolic regression requiring a hand‑crafted library;
> - LatentSDE [1]: neural model trained from scratch per data set;
> - FIM‑SDE: pre‑trained checkpoint, optionally fine‑tuned.
>
> The task is to infer the SDE that best characterizes the datasets.
>
> *Trade-offs*:
>
> | Method               | Prior knowledge required            | Training iterations | MMD quality |
> |----------------------|-------------------------------------|---------------------|-------------|
> | BISDE                | Hand‑crafted basis for each system  | $0$ (effectively)   | good        |
> | LatentSDE            | none                                | $5\\ 000$ (typical) | good        |
> | FIM-SDE (zero‑shot)  | none                                | $0$                 | good        |
> | FIM-SDE (fine‑tuned) | none                                | $\\mathbf{100}$     | **best**    |
>
> By not relying on prior knowledge and already achieving strong zero-shot performance, FIM‑SDE is the most suitable choice when compute time is limited or unavailable. When fine-tuning is permitted, FIM‑SDE further improves and consistently outperforms both baselines across all systems, requiring only $100$ epochs to reach state-of-the-art quality.
>
> *Details on convergence:  LatentSDE vs FIM-SDE fine-tuning*: The table below compares the performance trajectories of LatentSDE and FIM‑SDE over the course of training. LatentSDE requires thousands of iterations to converge, while FIM‑SDE reaches comparable or superior performance after only $100$ fine-tuning epochs—representing a $\\, > 50 \times$ reduction in compute. This highlights the efficiency of leveraging a pre-trained FIM‑SDE model.
>
>
>
> | Model                            | Wind               | Oil                | Facebook           | Tesla                |
> |:---------------------------------|:------------------:|:------------------:|:------------------:|:--------------------:|
> | LatentSDE (Epoch 500)            | $1.019 \\pm 0.078$ | $0.382 \\pm 0.236$ | $0.091 \\pm 0.043$ | $0.066 \\pm 0.059$   |
> | LatentSDE (Epoch 1000)           | $0.874 \\pm 0.127$ | $0.224 \\pm 0.185$ | $0.071 \\pm 0.037$ | $0.052 \\pm 0.056$   |
> | LatentSDE (Epoch 2000)           | $0.318 \\pm 0.164$ | $0.205 \\pm 0.172$ | $0.034 \\pm 0.035$ | $0.023 \\pm 0.028$   |
> | LatentSDE (Epoch 5000)           | $0.350 \\pm 0.212$ | $0.235 \\pm 0.167$ | $0.014 \\pm 0.018$ | $0.016 \\pm 0.023$   |
> |                                  |                    |                    |                    |                      |
> | FIM-SDE (zero-shot)              | $0.330 \\pm 0.11$4 | $0.199 \\pm 0.137$ | $0.012 \\pm 0.026$ | $0.000 \\pm 0.000$   |
> | FIM-SDE (fine-tuned) (Epoch 20)  | $0.114 \\pm 0.06$9 | $0.071 \\pm 0.094$ | $0.020 \\pm 0.019$ | $0.021 \\pm 0.014$   |
> | FIM-SDE (fine-tuned) (Epoch 40)  | $0.075 \\pm 0.05$2 | $0.048 \\pm 0.057$ | $0.019 \\pm 0.026$ | $0.020 \\pm 0.028$   |
> | FIM-SDE (fine-tuned) (Epoch 60)  | $0.042 \\pm 0.04$4 | $0.049 \\pm 0.034$ | $0.003 \\pm 0.007$ | $0.013 \\pm 0.018$   |
> | FIM-SDE (fine-tuned) (Epoch 80)  | $0.043 \\pm 0.07$7 | $0.046 \\pm 0.028$ | $0.002 \\pm 0.005$ | $0.011 \\pm 0.016$   |
> | FIM-SDE (fine-tuned) (Epoch 100) | $0.029 \\pm 0.03$4 | $0.068 \\pm 0.042$ | $0.006 \\pm 0.007$ | $0.004 \\pm 0.008$   |
>
>
> These results illustrate a clear use-case for FIM‑SDE: when facing a new real‑world system, one can immediately deploy the zero‑shot model without any training, obtaining results competitive with fully trained baselines. If better accuracy is needed and minimal compute is available, fine-tuning FIM‑SDE for just $100$ iterations suffices to outperform both symbolic and neural methods trained from scratch. This flexibility—in balancing training cost and regression quality—sets FIM‑SDE apart from on-the-go approaches, and makes it a practical foundation model for scientific time series.
>
> **[PROPOSED MODIFICATION:]** We will add all this new data to the Appendix.
>
> *References:*
>
> [1] X. Li, et al. Scalable Gradients for Stochastic Differential Equations. AISTATS 2020.
>
> [2] Bartosh et al. SDE Matching: Scalable and Simulation-Free Training of Latent Stochastic Differential Equations. ICML 2025
>
> [3] Wang et al. Data-driven discovery of stochastic differential equations. Engineering, 2022.

---

> > ### Comment · Reviewer_P9Vv · 2025-08-04
> >
> > I thank the author for the detailed explanations and new experimental results. I will remain my positive feedback.

---

### Official Review · Reviewer_np9z · 2025-06-28

**Clarity:** 4
**Significance:** 4
**Originality:** 3
**Rating:** 5
**Confidence:** 3

**Summary:**

In this paper, the authors introduce FIM-SDE for estimating the underlying dynamics of SDEs from noisy time series data. This is a pretrained recognition model that performs in-context (zero-shot) inference of both drift and diffusion functions without requiring fine-tuning or prior knowledge of the system. A notable strength of the method is it can learn from many simulated examples and works across a wide range of systems, from physical models to real-world data like stock prices and wind speeds. It's a flexible and general tool for understanding complex, noisy dynamics.

**Questions:**

- Regarding the data generation process described in Lines 152–153, could the authors clarify whether the use of randomly sampled dynamics truly reflects real-world scenarios? From my perspective, many sampled dynamics may lack physical interpretability or relevance.
- Regarding the proposed FIM-SDE, I am curious about the validity of treating x as the spatial location. This implicitly reformulates the SDE as a PDE, but x here is not location of fixed grid as required by SDEs.

**Ethical Concerns:**

["NO or VERY MINOR ethics concerns only"]

**Final Justification:**

All of my questions have been addressed by the authors. I think this work is solid and well-executed, and thus I recommend accept.

**Limitations:**

- The related work on Neural SDEs appears to be outdated and insufficient. The authors should incorporate discussions of more recent work in Neural SDEs and clarify how FIM-SDE differs from these approaches.
- The test data are limited to quite low-dimensional systems (at most 3), leaving the scalability of the method on higher-dimensional dynamics underexplored.

**Quality:**

3

**Strengths And Weaknesses:**

- This work investigates the problem of identifying latent SDEs from noisy observations. This is a very appealing and practical challenge. The contributions in both dataset construction and model design are noteworthy and appreciated.
- The proposed FIM-SDE model introduces a somewhat novel approach to SDE inference by leveraging amortized learning and neural operator principles.
- The experiments are sufficient and convincing, verifying the superiority of the proposed method.

---

> ### Author Rebuttal · Authors · 2025-07-30
>
> We thank the reviewer for their encouraging feedback, as well as for highlighting potential limitations and raising thoughtful questions. Below, we address each of the points.
>
> **@Q1**: One of our core assumptions is that the class of SDEs with polynomial drift, of degree at most three, and polynomial diffusion, of degree at most two, already cover many models used in the natural and social sciences. All $8$ canonical systems in Section 4.1 fall inside this class.
>
> Whether such randomly sampled SDEs reflect real-world scenarios can be judged from our real‑world evaluations. Below we reproduce Table 2.a and add
>
> (i) LatentSDE [1] (a strong neural network baseline), and
>
> (ii) FIM-SDE after fine-tuning on each dataset
>
> | Model           | Wind               | Oil                | Facebook           | Tesla                |
> |:----------------|:------------------:|:------------------:|:------------------:|:--------------------:|
> | BISDE           | $0.000 \\pm 0.000$ | $0.137 \\pm 0.097$ | $0.053 \\pm 0.060$ | $0.013 \\pm 0.015$   |
> | LatentSDE       | $0.350 \\pm 0.212$ | $0.235 \\pm 0.167$ | $0.014 \\pm 0.018$ | $0.016 \\pm 0.023$   |
> | FIM-SDE (zero-shot) | $0.330 \\pm 0.114$ | $0.199 \\pm 0.137$ | $0.012 \\pm 0.026$ | $0.000 \\pm 0.000$   |
> | FIM-SDE (fine-tuned)   | $0.029 \\pm 0.034$ | $0.068 \\pm 0.042$ | $0.006 \\pm 0.007$ | $0.004 \\pm 0.008$   |
>
> *Key take‑aways*:
>
> - *In zero‑shot mode*, FIM‑SDE is already competitive with both symbolic (BISDE) and neural (LatentSDE) methods, suggesting that pre‑training on our randomly sampled SDEs captures useful structure.
> - The structure provides a good basis for inference via finetuning. A brief fine‑tune via Eq. (12) (App. D) lets FIM‑SDE outperform both baselines on all four real‑world systems.
>
> **[PROPOSED MODIFICATION:]** We will include the new results into Section 4.
>
> **@Q2**: We may be misunderstanding the concern, so we kindly ask for clarification. The variable $\\mathbf{x}$ in Eq. 1 denotes *the state of the system*. FIM-SDE implements a map from:
>
> (i) (the space of) noisy observations on sampled paths drawn from SDEs like Eq. 1; to
>
> (ii) (the space of) *state-dependent* drift and diffusion functions that govern those paths.
>
> No PDE reformulation is involved.
>
> **@Limitation1**: We thank the reviewer for this observation. Appendix A provides a more detailed discussion of related work than the overview in the Introduction. There, we included six recent neural SDE models which, to the best of our knowledge, cover the most recent advances in neural SDE modeling.
>
> **[PROPOSED MODIFICATION:]** To make this discussion more visible we will
>
> 1. create a dedicated Related Work section in the main paper, and
>
> 2. move the discussion of the latest neural‑SDE papers from the appendix into that section, explicitly contrasting their objectives and assumptions with ours.
>
> **@Limitation2**: We acknowledge this limitation. As stated in both our abstract and introduction (line 79), the present work **targets low‑dimensional SDEs (1 to 3 dimensions) only**. Section 4 (lines 240–241) and the conclusion (lines 327–332) both reiterate this scope and outline the challenges of extending to higher‑D systems.
>
> In the reply **@W2** to Reviewer Kq3M we explain one of the main reasons why we, in this first study, restrict our attention to low-dimensional SDEs. We plan to address this in future work.
>
> *References*:
>
> [1] X. Li, et al. Scalable Gradients for Stochastic Differential Equations. AISTATS 2020.

---

> > ### Comment · Reviewer_np9z · 2025-08-05
> >
> > Thank you very much for your efforts during the rebuttal stage. I believe all of my concerns have been adequately addressed. I am particularly impressed by the strong generalization performance of the neural network simulators trained on synthetic data, and seemingly this is a common phenomenon. Thank you again for your detailed responses. I have accordingly raised my score to accept.

---

### Official Review · Reviewer_WGed · 2025-07-02

**Clarity:** 3
**Significance:** 1
**Originality:** 4
**Rating:** 4
**Confidence:** 3

**Summary:**

The paper introduces FIM-SDE, a neural network model designed for in-context learning of stochastic differential equations (SDEs). Its main contributions are
1. Developing a pretrained neural method capable of estimating drift and diffusion functions directly from noisy data across diverse tasks.
2. Framing the SDE discovery task as an in-context inference problem, enabling zero-shot generalization to different SDE systems.
3. Demonstrating that FIM-SDE can handle noisy, sparse, and irregularly sampled data effectively, outperforming traditional approaches.
4. Providing a unified architecture that maintains consistent representations of drift and diffusion functions across various datasets.
Overall, it advances the state of data-driven SDE discovery by combining neural network amortization, in-context learning, and distributional inference techniques.

**Questions:**

1. What exactly does $\Delta y^2$ represent? If $\Delta y$ already carries the full information, what is the purpose of introducing $\Delta y^2$?

2. The input consists of K(L−1) tuples. Is it provided as a sequence, similar to the standard Transformer input format? If only fewer tuples are available, would the model still work?

3. The paper focuses only on diagonal diffusion. Is there a specific reason or limitation behind this choice? Additionally, are there any other constraints on the diffusion term, such as requiring it to be invertible? If invertibility is not enforced, does the SDE formulation fully encompass ODEs as a special case? And how well does the model perform on tasks governed by deterministic dynamics?

4. Is there any restriction on the step size? What is the upper bound, if any? In general, the allowable step size is often related to the stiffness of the equation. For this type of SDE foundation model, how are different scales—both large and small—handled in practice?

5. Building on the previous question, suppose an SDE from the training set has its drift term scaled by a large coefficient. Would the resulting SDE, potentially exhibiting significantly different dynamics, still be accurately learned by the model?

6. How does the proposed method compare to neural network approaches that learn a specific SDE when evaluated on a given single test case? What is the magnitude of the performance difference？

**Ethical Concerns:**

["NO or VERY MINOR ethics concerns only"]

**Final Justification:**

The detailed response, as well as the additional experiments provided, has addressed my concerns.

**Limitations:**

Yes

**Quality:**

4

**Strengths And Weaknesses:**

Strengths:

1. The proposed novel FIM-SDE model leverages in-context learning, enabling it to infer drift and diffusion functions directly from raw data without retraining for each new system. This approach significantly enhances flexibility and efficiency in SDE discovery.

2. FIM-SDE demonstrates robustness to noisy, irregular, and sparsely sampled data, which are common in real-world scenarios.

3. The model can perform inference rapidly in a single forward pass.

Weakness:
As with many neural approaches, formal guarantees on approximation accuracy and convergence are limited. That said, the work still lacks a clear explanation of why only a few demonstrations suffice to train the SDE foundation model.

---

> ### Author Rebuttal · Authors · 2025-07-30
>
> We thank the reviewer for their detailed review and insightful comments, which have helped us improve the presentation of our manuscript. Below, we address each of the points raised.
>
> **@W1:** To answer this point precisely we would appreciate a brief clarification: does the reviewer mean by demonstrations
>
> (i) the number of context tuples per SDE instance, or
>
> (ii) the number of distinct SDEs used during pre‑training?
>
> **@Q1:** Both Gaussian‑process regression and symbolic regression recover drift and diffusion by exploiting Eqs. 2 and 3:
>
> - the expectation of $\\Delta y$ is proportional to the local drift,
> - the expectation of $\\Delta y^2$ is proportional to the local diffusion.
>
> We therefore explicitly provide both $\\Delta y$ and $\\Delta y^2$ as inputs to FIM‑SDE, allowing the network to learn separate feature channels for drift and diffusion estimation.
>
> **@Q2**: As noted in lines 189-193, we *assume* the sequential nature of the time series does not provide additional information for the *local* estimation of drift and diffusion, beyond that encoded into the one-step transitions ($\\Delta y$, $\\Delta y^2$). Consequently, FIM‑SDE treats each of the $K(L-1)$ tuples as iid variables (i.e. no positional encodings are used).
>
> *Number of tuples required by FIM-SDE:* During pre‑training, we randomly subsample between $128$ and $12\\ 800$ tuples per SDE, so the model is exposed to widely varying context sizes. In our experiments, the $12$ target data sets range from $5\\ 000$ tuples (canonical systems) to $20\\ 000$ tuples (stock data). FIM-SDE nicely generalizes to every case.
>
> **[PROPOSED MODIFICATION:]** We thank the reviewer for making us realize that this information was missing from the Appendix. We will include an appendix table that summarises context‑size statistics for all training and target sets.
>
> Finally, we have additionally revisited the performance of FIM-SDE on the double well (Eq. 24) and damped linear (E. 25-26) problems when varying the context size (in tuples). We considered the clean and dense observation case ($\\rho=0, \\Delta \\tau = 0.002$). The table below shows that FIM-SDE can correctly infer the damped linear oscillator with only $2\\ 000$ tuples. The double well problem, albeit 1D, is more complex because it requires characterising a state-dependent diffusion. FIM-SDE requires at least $4\\ 000$ tuples to correctly capture this structure.
>
> | Context Size | Double Well (MMD)   | Damped Linear (MMD) |
> |-------------:|:-------------------:|:-------------------:|
> | $500     $   | $0.155 \\pm 0.090 $  | $ 0.128 \\pm 0.068 $ |
> | $1\\ 000  $   | $0.115 \\pm 0.094 $  | $ 0.014 \\pm 0.005 $ |
> | $2\\ 000  $   | $0.053 \\pm 0.073 $  | $ 0.006 \\pm 0.005 $ |
> | $3\\ 000  $   | $0.042 \\pm 0.086 $  | $ 0.002 \\pm 0.003 $ |
> | $4\\ 000  $   | $0.004 \\pm 0.003 $  | $ 0.000 \\pm 0.000 $ |
> | $5\\ 000  $   | $0.004 \\pm 0.003 $  | $ 0.000 \\pm 0.000 $ |
> | $50\\ 000 $   | $0.006 \\pm 0.006 $  | $ 0.000 \\pm 0.000 $ |
>
>
>
> These results showcase the FIM-SDE’s ability to handle a wide range of context sizes, and will also be included into the Appendix.
>
> **@Q3.a (Is there a specific reason or limitation behind this choice?)**: For this first study we opted for diagonal diffusion to limit the number of functions to infer. Nothing in the architecture prevents non‑diagonal diffusion, and extending pre‑training to that setting is straightforward. We leave this extension for future work. Let us remark, nevertheless, that diagonal and state-dependent diffusion matrices represent a rich family of SDEs with wide application in the natural sciences. What is more, this assumption is consistent with all baseline models.
>
> **@Q3.b (are there any other constraints on the diffusion term?)**: No additional constraints are imposed beyond the polynomial diffusion of degree $\\leq 2$ prior used in pre-training (Section 3.1, lines 157-163).
>
> **@Q3.c (does the SDE formulation fully encompass ODEs as a special case?)**: Yes. The training corpus includes SDEs with **constant or zero diffusion**, so FIM‑SDE has seen purely deterministic trajectories. Nonetheless, because most training cases have non‑zero diffusion, the model is biased towards inferring non‑constant diffusion. The results we report in **@Q6** below illustrate this bias. They also demonstrate that the bias can be corrected via fine-tuning.
>
> **@Q4**: Each target data set has its own spatial and temporal scales, so all inputs are normalised as described in Section 3.2 (lines 187–188) and Appendix E.1. During data generation we simulated our sampled SDEs on a fine grid $\\Delta t = 10^{-3}$, but recorded observations at three spacings: $\\Delta \\tau \in \\{10^{-1}, 10^{-4}, 10^{-3}\\}$. Pretraining FIM-SDE on such paths teaches it to handle both sparse and dense observations.
>
> Notably, in our target datasets $\\Delta \\tau$ ranges from $0.2$ (wind speeds) to $2 \\times 10^{-5}$ (stock data) and FIM-SDE is able to handle all systems across that span.
>
> **[PROPOSED MODIFICATION:]** We again thank the reviewer for pointing out that this information was missing. We will include an appendix table that reports the inter-observation time scales for training and target systems.
>
> **@Q5**:  Scaling the drift can be viewed (e.g. through an Itô change of variables) as scaling the diffusion downwards. As explained in our reply **@Q3.c**, the pre-training distribution already contains vanishing‑diffusion cases, so the model remains capable of fitting such systems. Fine‑tuning on a handful of trajectories further removes any residual bias toward non‑zero diffusion (see **@Q6** below).
>
> **@Q6**: Comparison with neural network approaches.
>
> *Baseline.* LatentSDE [1] is a widely adopted neural model for SDE inference, so we use it as the task‑specific baseline.
>
> *Experimental setup.* We follow [1] exactly and consider the stochastic Lorenz attractor with parameters $(\\sigma, \\rho, \\beta)=(10, 28, 8/3)$ and a diagonal diffusion matrix with scale $0.15$. LatentSDE is trained on $1\\ 024$ paths $\\times\\ 41$ time points, sampled from the initial condition $\\mathcal{N}(0, 1)$. FIM‑SDE sees the same data as context. We evaluate it (i) zero‑shot and (ii) after fine‑tuning on those paths.
>
> *Evaluation setup.* For each of three initial‑condition sets: $\\{\\mathcal{N}(0, 1), \\mathcal{N}(0, 2), (1, 1, 1)\\}$, we generate ground‑truth trajectories and compute the Maximum Mean Discrepancy (MMD) to samples from each model.
>
> *Results*:
>
> | Model         | $\\mathcal{N}(0,1)$ | $(1, 1, 1)$ | $\\mathcal{N}(0, 2)$ |
> |:--------------|:------------------:|:-----------:|:-------------------:|
> | LatentSDE     |  $0.051708$        |  $0.135003$ |  $0.054335$         |
> | FIM-SDE (zero-shot) |  $0.403933$        |  $1.151080$ |  $0.378515$         |
> | FIM-SDE (fine-tuned) |  $0.003838$        |  $0.037159$ |  $0.003335$         |
>
> LatentSDE clearly outperforms FIM‑SDE when the latter is evaluated zero‑shot. After fine‑tuning for a few hundred epochs, however, FIM‑SDE surpasses LatentSDE on all three tests.
>
> Let us remark that FIM‑SDE is only fine-tuned on the training set of LatentSDE — which was sampled from the initial condition $\\mathcal{N}(0, 1)$. Paths sampled from the other two initial conditions are only used during model evaluations.
>
> **Why is the zero‑shot performance weaker?**
>
> The data are drift‑dominated. We invite the reviewer to quickly check Fig 6 in the LatentSDE paper [1]. Because our pre‑training set contains mostly non‑zero diffusion functions, FIM‑SDE is biased toward inferring finite diffusion, which increases the variance of its generated paths that lead to larger MMD scores. We will add a diagnostic plot of the zero-shot inferred diffusion illustrating this bias to the Appendix.
> Fine‑tuning on the task data quickly removes the bias and restores accuracy.
>
> **Convergence speed**
>
> A recent benchmark, SDEmatching [2], which studied the same stochastic Lorentz attractor setup,
> shows that LatentSDE typically converges only after ${\\approx}\\ 5\\ 000$ iterations (we have verified this), whereas SDEmatching needs ${\\approx}\\ 500$.
>
> Next, we compare those convergence speeds against (i) fine‑tuning FIM‑SDE and (ii) training FIM‑SDE from scratch:
>
> |    FIM Training Mode   | It.  50     | It.  100    | It.  200    | It.  500    | It.  1 000 | It.  2 000 | It.  5 000 |
> |:-------------------|:------------:|:------------:|:------------:|:------------:|:------------:|:------------:|:------------:|
> |      Fine-tune      | $0.012888$   | $0.003838$   | $0.003247$   | $0.000000$   | $0.0000000$  | $0.0000000$  | $0.0000000$  |
> | Train from scratch | $0.749462$   | $0.491508$   | $0.397677$   | $0.203864$   | $0.0982177$  | $0.0154879$  | $0.0125062$  |
>
>
> *Key take‑aways*
> - Fine‑tuning FIM‑SDE reaches near‑optimal performance within ${\\leq}\\ 100$ epochs — much faster than both LatentSDE and SDEmatching.
> - Pre‑training provides an excellent initialisation; training from scratch requires two orders of magnitude more iterations to match the same error.
>
> We also invite the reviewer to check our response **@Q3** for reviewer Kq3M, where we present a comparison between FIM-SDE and LatentSDE on the real-world datasets.
>
> **[PROPOSED MODIFICATION:]** All these results will be inserted into the manuscript.
>
> **NOTE ON TABLE 2B (paper) vs TABLE ABOVE**
>
> The results in the first table above differ from those in Table 2b of the main text, as the latter were computed using an incorrect normalization factor. However, the corrected results presented in this rebuttal can be easily reproduced using the pretrained FIM-SDE model provided with our submission.
>
> *References*:
>
> [1] X. Li, et al. Scalable Gradients for Stochastic Differential Equations. AISTATS 2020.
>
> [2] Bartosh et al. SDE Matching: Scalable and Simulation-Free Training of Latent Stochastic Differential Equations. ICML 2025

---

> > ### Comment · Reviewer_WGed · 2025-08-04
> >
> > Thank you very much for the detailed response, as well as the additional experiments provided. The reply has addressed my concerns. Assuming all these are incorporated in the revised version, I will raise my score.

---

### Official Review · Reviewer_Kq3M · 2025-07-03

**Clarity:** 3
**Significance:** 2
**Originality:** 3
**Rating:** 4
**Confidence:** 4

**Summary:**

This paper seeks to pre-train a large foundation model on the supervised mapping of SDE realizations to their associated drift and diffusion coefficients so that later on, as an alternative to GPs and neural SDEs.

The authors generate a dataset of 600k low-dimensional SDEs, make their observations a bit noisy, and learn to regress the associated drift and diffusion coefficients giving rise to them.

They then show that the tool can be used to predict the coefficients of unseen SDEs that fall within the class, and compare to existing methods.

**Questions:**

- The reviewer is trying to understand how fair the comparison is to GPs. GPs also give you a nice measure of uncertainty on your output. Can you get the same from your approach, rather than just a pointwise mapping?

- Can the authors comment more on why they only chose this sparse set of low dimensional SDEs? Is it the difficulty trying to come up with higher dimensional dataset?

- The "generalization" performance that the authors discuss for zero-shot prediction on real world datasets seems not strongly well founded. Surely there are more than four stochastic systems of real data to test on, and matching the performance of one other method is not the strongest argument for generalization. Could the authors investigate this further for the rebuttal? This is a pretty limited set of experiments to justify one of the main interesting claims of the paper.

**Ethical Concerns:**

["NO or VERY MINOR ethics concerns only"]

**Final Justification:**

The authors provided more experimental justification of their method on real-world datasets, showing that their proposed training approach was expansive enough to encompass meaningfully interesting SDEs and not just proof-of-principle ones. Nonetheless, they also helped make clear that there is a curse of dimensionality in terms of number of training points needed to generalize that is pretty steep, so I'm keeping my score at a 4 (weak accept).

**Limitations:**

yes

**Quality:**

3

**Strengths And Weaknesses:**

*Strengths*

- While the paper relies on a very constrained parametric class of drift and diffusion coefficients on which the synthetic datasets are constructed, they are very upfront on this, and the reviewer sees this paper more as a proof of concept on building these foundation models
- The phrasing of the approach is interesting and seems novel as I understand it, and, with the right generalization of function classes and datasets, could be an interesting approach going forward.

*Weaknesses*

- My main criticism is something the authors will likely agree with, namely that the generalization of the "foundation model" to SDEs that it has not been trained on makes sense when the tested models are still within, in some sense, the parametric class of polynomial drifts of degree 3. One ablation that should be done but that I don't see in the paper is if the test SDEs have drift and diffusions that fall within the class of polynomials of degree 3.
- My other main criticism is the low dimensionality of the target SDEs. Why is this the case? A computational limitation? It does not seem absolutely necessary to the reader that they proceed this way, as there exist many paradigms for learning high dimensional SDEs already (e.g. score-based diffusion). Is the issue coming up with interesting candidate training datasets?
- Equation 8 is not well explained. You do not address why you are introducing $U$ other than that it can be interpreted as a measure of epistemic uncertainty and a citation to another paper. This needs to be better motivated.




*Minor comments*
- typo with unnecessary comma line 57
- MSE is not a divergence measure, you should consider changing this verbiage (line 215)

---

> ### Author Rebuttal · Authors · 2025-07-30
>
> Before we begin, we would like to thank the reviewer for their detailed feedback, for identifying potential weaknesses and raising important questions, and for recognizing the contributions of our work. We address each of the raised points below.
>
> **@W1**:  We agree with the reviewer that FIM‑SDE can be expected to generalize only to *(unseen) SDEs that are within the SDE class used to build the synthetic training set of Section 3.1*. However, the practical impact of this limitation depends on whether that class is already rich enough to cover SDEs used in real-world applications.
>
> FIM‑SDE was trained on data drawn from the class
>
>
> $\\mathcal{C} = \\{\\text{SDEs with polynomial drift of degree} \\leq 3, \\text{ and polynomial diffusion of degree} \\leq 2\\}$,
>
> where by diffusion we refer to the $\\{g_1, …, g_d\\}$ functions in Eq. 4.
>
> *Why $\\mathcal{C}$ is already practically useful*:
>
> - The eight benchmark SDEs used in Section 4.1 all belong to $\\mathcal{C}$. The reviewer can readily check this by looking at their definition in Appendix G.4.  These SDEs are widely used across natural and social sciences alike.
> - In Section 4.2, we studied four widely different real-word datasets, namely Facebook and Tesla stock prices, and oil price and wind speed fluctuations.  Stock prices and oil price fluctuations are traditionally modelled as geometric Brownian motion (linear drift and diffusion, see e.g. [1, 2] for the latter), while wind speed fluctuations have been recently described with linear drift and quadratic diffusion [3]. All these models belong to $\\mathcal{C}$.
> - Pretraining FIM‑SDE on data sampled from $\\mathcal{C}$ provides a good initialization for fine-tuning the foundation model to data (See our replies **@Q3** to you, and **@Q6** to reviewer WGed below).
>
> In short, many SDEs of practical interest already satisfy the polynomial restrictions, and FIM‑SDE shows consistent generalization to them.
>
> **[PROPOSED MODIFICATION:]** We will include the discussion above on generalization within class $\\mathcal{C}$ into our experimental section (Section 4).
>
> **[QUESTION TO REVIEWER:]** Does this address the missing ablation the reviewer requested? If further evidence is required, we can include an ablation on degree‑4 drift SDEs (outside $\\mathcal{C}$) and report the expected drop in performance.
>
> **@W2**: The intuition of the reviewer is correct. One of the main reasons why we, in this first study, restrict our attention to low-dimensional SDEs are the **data‑generation rejection rates**.
>
> As we described in lines 582-587 (Appendix C3), we simulated $100$ paths per candidate SDE and rejected any SDE whose paths contain NaNs, Infs, or values exceeding a threshold value of $\\pm 100$. We have empirically found that the rejection rate rises sharply with dimension: $\\mathbf{50\\%}$ **for 1‑D,** $\\mathbf{78\\%}$ **for 2‑D, and** $\mathbf{92\\%}$ **for 3‑D**. Higher dimensionality therefore demands far more SDE candidates to populate the training distribution, illustrating a *curse‑of‑dimensionality effect*.
>
> In future work, we plan to reduce the rejection rates of our algorithm by generating SDEs that are *a priori known to have a solution* (i.e. by randomly generating drift and diffusion functions that are globally Lipschitz and feature linear growth). Note that we already commented on this in lines 322-326 of our Conclusions.
>
> **[PROPOSED MODIFICATION:]** We will state this bottleneck more explicitly and include the rejection rate statistics in the Limitation Section of the manuscript.
>
> **@W3**: In lines 221-224 of our manuscript we explained that when one marginalises Eq. 7 over a uniform prior on $\\mathbf{x}$, the integrand can be dominated by regions where the signal‑to‑noise ratio is extremely high (or low), producing large gradients and numerical instability. In line 226 we then wrote that the learnable and state-dependent $U_\\theta$  function works as a weighting mechanism that is introduced to address these instabilities.
>
> *Interpretation as epistemic uncertainty*: Assuming that Eq. 7 is a state-dependent Gaussian variable, $U_\\theta$ plays the role of the learnable, state-dependent log-variance. If we look at Eq. 8, large $U_\\theta$ values down‑weight regions where the model is uncertain, while the additive $+U_\\theta$ term prevents it from collapsing to $-\\infty$. Empirically, we have observed that $U_\\theta$​ indeed peaks in sparsely observed or highly noisy areas, acting as a proxy for epistemic uncertainty.
>
> **[PROPOSED MODIFICATION:]** We will add this explanation to the main text of our manuscript and provide a plot of $U_\\theta$ for a representative example in the Appendix.
>
> **@Q1**:  The primary goal of our proposal is to perform SDE inference from noisy data *in a single forward pass*. In other words, FIM-SDE was trained to perform zero-shot pointwise mappings only. **However, uncertainty estimation is a natural next step**. In fact, as we discussed in **@W3**, the function $U_{\\theta}$ can also be interpreted as an uncertainty estimator *wrt. the loss function defined in Eq. 7*. Future work will consider two independent $U$ functions, one that weights the drift contribution (first term  in Eq. 7), and the other that weights the diffusion contribution (second term in Eq. 7). Each of these $U$ functions can then be used as uncertainty estimators for the drift and diffusion functions, respectively.
>
> **@Q2**: We kindly refer the reviewer to our reply **@W2** above.
>
> **@Q3**: In the $16$ papers that we reviewed in our related work, Appendix A, we unfortunately did not find many real-world datasets on which to test FIM-SDE, but the ones that we considered. In fact, *most works typically only test their proposals on synthetic datasets*.
>
> Having said this, we have now extended our experiments on the $4$ real-world experiments, Table 2a of the manuscript, by including the scores of
>
> (i) LatentSDE [5] (one of the most prominent neural network models for SDE inference); and
> (ii) FIM-SDE *fine-tuned* to the training sets.
>
> | Model           | Wind              | Oil               | Facebook          | Tesla               |
> |:----------------|:-----------------:|:-----------------:|:-----------------:|:-------------------:|
> | BISDE           | $0.000 \pm 0.000$ | $0.137 \pm 0.097$ | $0.053 \pm 0.060$ | $0.013 \pm 0.015$   |
> | LatentSDE       | $0.350 \pm 0.212$ | $0.235 \pm 0.167$ | $0.014 \pm 0.018$ | $0.016 \pm 0.023$   |
> | FIM (zero-shot) | $0.330 \pm 0.114$ | $0.199 \pm 0.137$ | $0.012 \pm 0.026$ | $0.000 \pm 0.000$   |
> | FIM (fine-tuned)   | $0.029 \pm 0.034$ | $0.068 \pm 0.042$ | $0.006 \pm 0.007$ | $0.004 \pm 0.008$   |
>
> *Key take‑aways*:
>
> - The **zero-shot** performance of FIM-SDE on real-world data is competitive with *both* symbolic regression methods (BISDE) and neural models (LatentSDE). These results highlight the robustness of FIM-SDE’s generalization across markedly different dynamical systems.
> - Leveraging Eq. 12 (Appendix D), FIM-SDE can be readily fine-tuned to real-world data. The fine-tuned model outperforms both baselines across all systems.
>
> These findings demonstrate that FIM-SDE functions as a true foundation model, exhibiting competitive zero-shot performance and the ability to be fine-tuned on complex data.
>
> **[PROPOSED MODIFICATION:]** We will include these new results into our experimental section.
>
> **@minor comments**: We thank the reviewer for pointing these issues out. We have corrected them in our manuscript.
>
> *References*:
>
> [1] Noh, N. M. et al. Analysis of oil price fluctuations. In AIP Conference Proceedings, volume 1750, 2016.
>
> [2] X. Yang and Z. He. Predicting the price of crude oil based on the stochastic dynamics learning from prior data. Stochastic Environmental Research and Risk Assessment, 2024.
>
> [3] Wang et al. Data-driven discovery of stochastic differential equations. Engineering, 2022.
>
> [4] d’Ascoli et al.  ODE-Former: Symbolic regression of dynamical systems with transformer. ICLR, 2024.
>
> [5] X. Li, et al. Scalable Gradients for Stochastic Differential Equations. AISTATS 2020.

---

> > ### Comment · Reviewer_Kq3M · 2025-08-04
> > **Follow-up**
> >
> > Thanks for your response and clarification of limitations and applicability. I am happy to keep my score of marginal accept!

---

### Decision · Program_Chairs · 2025-09-17

**Decision:**

Accept (poster)

**Comment:**

This paper introduces FIM-SDE, a framework for estimating the dynamics of stochastic differential equations from noisy time series data. The authors successfully pre-train a model to map SDE realizations to their drift and diffusion coefficients, allowing for zero-shot generalization to unseen SDEs. They generate a dataset of 600k low-dimensional SDEs and demonstrate the tool's effectiveness compared to existing methods.

All reviewers note that the authors have effectively addressed previous concerns. The strengths of the paper include its novel approach to in-context learning, robustness to noisy data, and rapid inference capabilities. While some limitations remain, such as the focus on low-dimensional systems and the need for clearer explanations of certain equations, the overall contributions are significant. The reviewers unanimously recommend acceptance, acknowledging the paper's potential to advance the field of SDE discovery.